# Effective Model Pruning: Measuring the Redundancy of Model Components

Yixuan Wang [1]   Dan P. Guralnik [2]   Saiedeh Akbari [1]   Warren E. Dixon [1]

## Abstract

This article initiates the study of a basic question about model pruning. Given a vector $s$ of importance scores assigned to model components, how many of the scored components could be discarded without sacrificing performance? We propose Effective Model Pruning (EMP), which derives the desired sparsity directly from the score distribution using the notion of effective sample size from particle filtering, also known as the inverse Simpson index. Rather than prescribe a pruning criterion, EMP supplies a universal adaptive threshold derived from the distribution of the score $s$ over the model components: EMP maps $s$ to a number $N_{eff} = N_{eff}(s)$, called the effective sample size. The $N - N_{eff}$ lowest scoring components are discarded. A tight lower bound on the effective mass $s_{eff}$ (the sum of retained normalized scores) in terms of $N_{eff}$ is derived. This process yields models with a provable upper bound on the loss change relative to the original dense model. Numerical experiments are performed demonstrating this phenomenon across a variety of network architectures including MLPs, CNNs, Transformers, LLMs, and KAN. It is also shown that EMP addresses a rich set of pruning criteria such as weight magnitude, attention score, KAN importance score, and even feature-level signals such as image pixels.

## 1. Introduction

Deep Neural Networks achieve remarkable results across numerous domains such as computer vision (Krizhevsky et al., 2012; He et al., 2016; Dosovitskiy, 2020), natural language processing (Vaswani et al., 2017; Devlin et al., 2019), robotics (Zitkovich et al., 2023), and generative artificial intelligence (Ho et al., 2020), through the deployment of increasingly large and complex models. While this growth has led to more accurate and generalizable models, it also introduces significant deployment challenges on edge devices due to high demands on computation, memory, and energy. Lack of resources is particularly evident when deploying large language models (LLMs) (Touvron et al., 2023a;b; Bai et al., 2023) and other over-parameterized models (Liu et al., 2023) in latency-sensitive or resource-constrained environments.

To address deployment challenges of such models, pruning has emerged as a fundamental and widely studied technique (Frankle & Carbin, 2018; Han et al., 2015; Cheng et al., 2024). The pruning field developed a rich taxonomy, typically categorized along three dimensions: what to prune (unstructured weights (Han et al., 2015), structured filters/channels (Li et al., 2016; Liu et al., 2017), or attention heads (Michel et al., 2019; Voita et al., 2019)), when to prune (before (Lee et al., 2018; Wang et al., 2020), during (Louizos et al., 2017; Evci et al., 2020), or after training (Frantar & Alistarh, 2023; Sun et al., 2023)), and how to score parameters (e.g., by magnitude (Han et al., 2015), sensitivity (LeCun et al., 1989; Hassibi et al., 1993), or data-driven metrics (Molchanov et al., 2016)). Despite extensive research in model pruning, a critical and persistent question remains: **given a neural network and the corresponding score vector $s$ derived from a pruning criterion, how many candidates should be retained?**

The choice of sparsity budget is sensitive. An overly aggressive budget degrades model performance, while an overly conservative one forfeits potential efficiency gains. Current solutions remain unsatisfactory, as sparsity often relies on expensive iterative pruning procedures (Renda et al., 2020), manual or heuristic per-layer budgets, or hyper-parameters that require careful tuning (Gale et al., 2019; Frantar & Alistarh, 2023). Recent work (Zhang et al., 2023) gives sharp lower and upper bounds on the pruning rate in terms of the specific loss change $\epsilon$ incurred by pruning.

Analogous sparsity selection problems exist in the setting of particle filtering (Kong et al., 1994; Doucet et al., 2001; Liu, 2009), ecology (Roswell et al., 2021) and statistical physics (Mezard & Montanari, 2009). In the particle filter setting, the notion of effective sample size (ESS) quantifying the number of statistically significant particles in a weighted

[1]Department of Mechanical and Aerospace Engineering, University of Florida [2]Department of Mathematics, Ohio University. Correspondence to: Yixuan Wang <wang.yixuan@ufl.edu>.

*Proceedings of the 43rd International Conference on Machine Learning*, Seoul, South Korea. PMLR 306, 2026. Copyright 2026 by the author(s).

set is widely used. ESS is a standard measure to evaluate degeneracy and guide resampling in sequential Monte Carlo methods. In ecology, the same measure is known as the inverse Simpson index. This literature draws on direct connections with Renyi entropies to produce measures of species diversity.

In this paper, we develop Effective Model Pruning (EMP) as a universal threshold determining retention directly from the score distribution. EMP is a simple rule that automatically determines the effective number $N_{eff} = N_{eff}(s)$ of top-scoring candidates to retain. This value $N_{eff}$ intuitively represents the number of truly significant contributors. By keeping the top $N_{eff}$ entries, EMP provides a simple computational criterion for deciding how many highest-scoring contributors to keep, in tandem with a tight theoretical lower bound on the retained mass, derived in Section 4.1.

EMP is a universal rule, agnostic of network architecture and pruning paradigm. It eliminates the need for manual budget scheduling and hyperparameter tuning, providing a versatile, robust and automatic pruning limit criterion. To validate the robustness of EMP, we examine the model's performance across a diverse range of criteria and network structures by pruning the model's entries by $\beta N_{eff}$ across a range of scaling coefficients $\beta$. Empirical results demonstrate that models pruned by EMP consistently achieve competitive performance with their dense counterparts, underscoring its effectiveness and generality.

Our contributions are as follows:

- We introduce Effective Model Pruning, a simple rule to convert any score vector $s$ into a principled sparsity threshold $N_{eff}$, supported by a theoretically guaranteed lower bound on the effective mass $s_{eff}$.

- We deduce an upper bound for the loss change $\epsilon$ between dense model and the sparse model obtained by EMP.

- We demonstrate the effectiveness of EMP across diverse architectures and pruning criteria, indicating it can be combined with existing criteria to achieve strong performance without additional tuning.

This article is organized as follows. Section 2 reviews related work on pruning criteria and targets. Section 3 discusses the geometry of the simplex in relation to $N_{eff}$. Section 4 develops the lower bound on effective mass $s_{eff}$, the loss change upper bound, and the EMP algorithm. Section 5 reports on the empirical study conducted to evaluate the role of $N_{eff}$ as a potential pruning threshold. The proof of Proposition 4.1 and the full experimental details appear in Appendices A and B. The code can be found at https://github.com/noMushroomw/Effective-model-pruning

## 2. Related Work

### 2.1. Pruning Criteria

Optimal Brain Damage (LeCun et al., 1989) and Optimal Brain Surgeon (Hassibi et al., 1993) estimate the loss increase caused by removing a parameter through second order approximations, and thereby prioritize removals that minimally perturb the objective. Magnitude based heuristics (Han et al., 2015) emerged as a simple and robust baseline in practice and were integrated into end to end compression pipelines that combine pruning with quantization and entropy coding. Empirical study (Gale et al., 2019) confirmed that magnitude-based criteria remain competitive across architectures when combined with careful scheduling and calibration.

Post-training pruning, which is particularly attractive for LLMs due to the prohibitive cost of retraining from scratch, has recently focused on simple but highly scalable criteria. SparseGPT (Frantar & Alistarh, 2023) performs one shot pruning with local least squares reconstruction to control the induced error in each block and achieves strong PPL at high sparsity without prolonged fine tuning. Wanda (Sun et al., 2023) introduces an activation-aware magnitude score that multiplies absolute weights by a norm of the corresponding activation statistics, thereby adapting the criterion to the data distribution seen at inference. These approaches retain the practical appeal of magnitude-based rules while injecting task awareness through reconstruction or activation weighting.

### 2.2. What to Prune

Unstructured pruning (Han et al., 2015; Gale et al., 2019) removes individual weights and maximizes flexibility in shaping sparsity patterns, while structured pruning removes entire computational units and thereby preserves dense tensor shapes that map efficiently to commodity accelerators. Representative methods in CNNs target filters (Li et al., 2016; He et al., 2019), channels (Luo et al., 2017), or neurons using criteria based on magnitude, batch normalization scaling factors (Liu et al., 2017), or Taylor approximations of the loss (Molchanov et al., 2016). In Transformer architectures, structured pruning often targets attention heads and intermediate feed forward channels. Empirical analyses showed that many heads are redundant for downstream tasks and can be excised with limited effect (Michel et al., 2019; Voita et al., 2019), while more recent large language model pipelines integrate structured removal of heads, MLP channels, or even layers with light recovery to obtain compact models amenable to further distillation or continued pretraining (Ma et al., 2023; Xia et al., 2023).

Semi-structured pruning strikes a compromise between irregular flexibility and hardware friendliness by enforcing lo-

cal patterns such as $N:M$-sparsity within rows or columns, which aligns with sparse tensor core primitives on modern GPUs. Learning and representing such patterns efficiently has been an active area of systems and algorithms research (Zhou et al., 2021; Castro et al., 2023). In practice, the choice among unstructured, structured, and semi-structured targets is driven by the deployment stack: when wall clock latency and throughput are paramount, structured or $N:M$-patterns commonly yield more predictable gains (Gale et al., 2019; Cheng et al., 2024).

## 3. Preliminary

In this section, we review the relationship between the sparsity and the model sharpness given by (Zhang et al., 2023)[Lemma 3.5], appearing here as Lemma 3.1.

Let $\hat{y} = f(\theta, x)$ denote a well-trained dense deep neural network with weights $\theta^* \in \mathbb{R}^N$ and empirical loss $L(\theta^*)$. A pruned network is derived from the dense network, whose weight is given by $\theta^k = \theta^* \odot M$, where $M$ is a binary mask matrix with $\|M\|_0 = k$ and $\odot$ is entrywise multiplication. Then the pruning ratio $\rho$ is defined as $\rho \triangleq k/N$.

Note that, due to the ever-growing sizes of modern learning architectures, it is especially interesting to consider asymptotics as $k, N \to \infty$ in regimes satisfying $k/N \to \rho \in (0, 1)$, allowing the study of pruning ratios for arbitrarily large networks. With this observation in mind, we cite—and later apply—the following result.

**Lemma 3.1.** *Given a well-trained neural network $f(\theta^*, x)$, let $\epsilon$ denote the loss difference, $|L(\theta^*) - L(\theta^k)|$, between the dense network and its pruned version, and let $H$ denote the Hessian matrix of the loss function $L$ with respect to the parameter, $\theta$. Then,*

$$\rho \leq 1 - \frac{2\epsilon N}{\|\theta^* - \theta^k\|_2^2 \operatorname{Tr}(H) + 2\epsilon N},$$

*where $\operatorname{Tr}(H)$ is the trace of the matrix $H$.*

Lemma 3.1 facilitates an upper bound on the loss change $\epsilon$ between the dense model and the EMP-pruned model, which is derived in Section 4.2.

### 3.1. Effective Sample Size and Geometry of the Simplex

Fix $N > 1$. Let $s \triangleq (s_1, \ldots, s_N)$ be a vector of scores associated with a pruning object. Define the normalized probability weight vector $\omega$ via

$$\omega_i \triangleq \frac{|s_i|}{\sum_i |s_i|}, \quad i = 1, \ldots, N. \tag{1}$$

The effective sample size $N_{eff} = N_{eff}(\omega)$ is defined as

$$N_{eff} \triangleq \left\lfloor \frac{1}{\sum_i \omega_i^2} \right\rfloor.$$

This section introduces a geometric interpretation of $N_{eff}$. Consider the standard $(N-1)$-simplex $\Delta$ in the Euclidean space $E = \mathbb{R}^N$ and the affine hyperplane $\Pi$ it spans:

$$\Delta \triangleq \left\{ \omega \in \mathbb{R}_{\geq 0}^N : \omega^\top \mathbf{1}_N = 1 \right\},$$
$$\Pi \triangleq \left\{ \omega \in \mathbb{R}^N : \omega^\top \mathbf{1}_N = 1 \right\}.$$

Thus, the vector $\omega$ constructed in equation 1 is a point of $\Delta$. Since both $\Delta$ and $N_{eff}$ are invariant under coordinate permutations, for any $\omega \in \Delta$ and $\nu \in [N]$, the coordinates can always be permuted so that the first $\nu$ coordinates are the largest $\nu$ weights. More precisely, if $\mathfrak{S}_N$ is the group of permutations on the set $[N] \triangleq \{1, \ldots, N\}$, then

$$\Delta = \bigcup_{\tau \in \mathfrak{S}_N} L_\sigma(\tilde{\Delta}),$$
$$\tilde{\Delta} \triangleq \{ \omega \in \Delta : \omega_1 \geq \omega_2 \geq \cdots \geq \omega_N \},$$

where $L_\tau : E \to E$ is the linear transformation satisfying $L_\tau(e_i) = e_{\tau(i)}$ for all $i \in [N]$. It follows that $L_\tau(\Delta) = \Delta$ and $N_{eff}(\omega) = N_{eff}(L_\tau \omega)$ for all $\tau \in \mathfrak{S}_N$ and all $\omega \in \Delta$. Note also that $L_\sigma(\tilde{\Delta})$ and $L_\tau(\tilde{\Delta})$ are geometric $(N-1)$-dimensional simplices with disjoint interiors whenever $\sigma, \tau \in \mathfrak{S}_N$ and $\sigma \neq \tau$.

Given $s$, compute $\omega = \omega(s)$ and find $\sigma \in \mathfrak{S}_N$ such that $L_\sigma(\omega) \in \tilde{\Delta}$. Then, the effective mass $s_{eff}$ is defined as

$$s_{eff} \triangleq \sum_{i=1}^{N_{eff}} \omega_{\sigma(i)}.$$

It follows that $s_{eff} = (L_\tau s)_{eff}$ for all $\tau \in \mathfrak{S}_N$, which makes it sufficient to study $s_{eff}$ restricted to $\tilde{\Delta}$, noting that

$$\omega \in \tilde{\Delta} \implies s_{eff} = \sum_{i=1}^{N_{eff}} \omega_i.$$

Letting

$$A_\nu \triangleq \left\{ \omega \in \tilde{\Delta} : \nu \leq \|\omega\|^{-2} < \nu + 1 \right\}, \tag{2}$$

one observes that $N_{eff}(\omega) = \nu$ if and only if $\omega \in A_\nu$. Computing a lower bound on $s_{eff}$ in terms of $N_{eff}$ is then tantamount to calculating the quantity $\inf_{\omega \in A_\nu} \varphi_\nu(\omega)$, where

$$\varphi_\nu(\omega) \triangleq \sum_{i=1}^{\nu} \omega_i = \nu \left\langle \omega \mid \zeta_{[\nu]} \right\rangle,$$

and where

$$\zeta_J \triangleq \frac{1}{|J|} \sum_{i \in J} e_i, \quad J \subseteq [N],$$

denotes the barycenter of the simplex $\operatorname{conv}(\{e_i : i \in J\})$. The challenge is that this optimization problem is not convex, due to the right-hand side inequality in equation 2.

## 4. Effective Model Pruning

From the identity

$$a, b \in \Delta \implies \left\langle a - \zeta_{[N]} \mid b - \zeta_{[N]} \right\rangle = \langle a \mid b \rangle - \frac{1}{N},$$

Illustration of the $B_\nu$ ball and the simplex $\Delta$

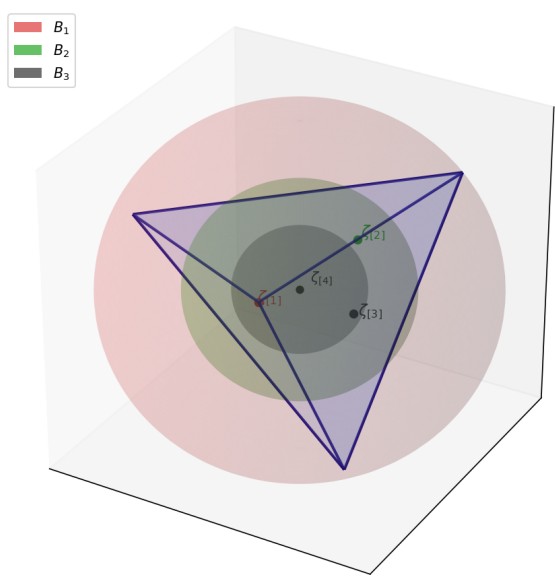

*Figure 1.* Illustration of the $B_\nu$ balls ($\nu = 1, 2, 3, 4$) and the simplex $\Delta$. Note that ball $B_4$ degenerates to the barycenter $\zeta_{[4]}$.

it follows that $A_\nu = \tilde{\Delta} \cap (B_\nu - B_{\nu+1})$, where

$$B_\nu \triangleq \left\{ \omega \in \Pi \colon \|\omega - \zeta_{[N]}\|^2 \leq \frac{1}{\nu} - \frac{1}{N} \right\},$$

for $\nu \in [N]$. Thus, $\varphi_\nu$ needs to be minimized over the intersection of $\tilde{\Delta}$ with the spherical shell in $\Pi$ obtained by subtracting the ball $B_{\nu+1}$ from the ball $B_\nu$. Note that $\zeta_{[N]}$ is both the barycenter of $\Delta$ and a vertex of $\tilde{\Delta}$. The vertices of $\tilde{\Delta}$ are precisely all the $\zeta_{[j]}$, $j \in [N]$, with $\zeta_{[1]}, \ldots, \zeta_{[\nu-1]}$ lying outside $B_\nu$, $\zeta_{[\nu]}$ lying on its boundary, and $\zeta_{[\nu+1]}, \ldots, \zeta_{[N]}$ lying in its interior. Each $B_\nu$ is a Euclidean ball in $\Pi$ of radius $r_\nu \triangleq \sqrt{\frac{1}{\nu} - \frac{1}{N}}$ about $\zeta_{[N]}$, and it is tangent at $\zeta_{[\nu]}$ to the $(\nu - 1)$-dimensional face of $\tilde{\Delta}$ given by $\mathrm{conv}(\zeta_{[\nu]}, \zeta_{[\nu-1]}, \ldots, \zeta_{[1]})$.

One must pay attention to the boundary cases, though. If $\nu = 1$, then $B_\nu$ contains all of $\tilde{\Delta}$ (and hence all of $\Delta$), while $B_{\nu+1} = B_2$ is the ball about $\zeta_{[N]}$ in $\Pi$ "caged" by the edges of $\Delta$. If $\nu = N$, then $B_\nu$ degenerates to a single point, $\zeta_{[N]}$. Finally, for $\nu = N - 1$, $B_\nu$ is the ball about $\zeta_{[N]}$ in $\Pi$, inscribed in $\Delta$, see Figure 1.

### 4.1. Lower Bound on the Effective Mass

With the observations of Section 3.1, a trivial lower bound on $s_{eff}$ is obtained by observing that

$$\inf_{\omega \in A_\nu} \varphi_\nu(\omega) \geq \inf_{\omega \in \tilde{\Delta}} \varphi_\nu(\omega) = \min_{i \in [N]} \nu \left\langle \zeta_{[i]} \mid \zeta_{[\nu]} \right\rangle$$

$$= \nu \left\langle \zeta_{[N]} \mid \zeta_{[\nu]} \right\rangle = \frac{\nu}{N},$$

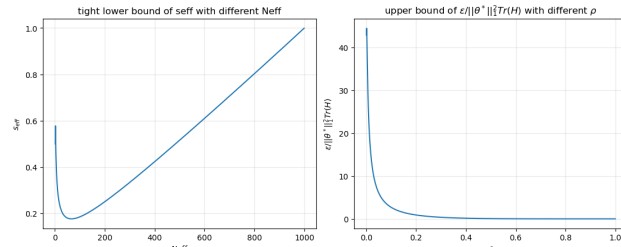

*Figure 2.* Lower and upper bounds associated with pruning. The left panel illustrates the tight lower bound of the effective mass $s_{eff}$ as a function of $N_{eff}$ for $N = 1000$. The right panel depicts the normalized upper bound of the loss change, $\epsilon / (\|\theta^*\|_1^2 \mathrm{Tr}(H))$, showing its rapid decay as $\rho$ increases.

since expanding the minimization domain to $\tilde{\Delta}$ makes the problem convex. In particular, $s_{eff} \geq \frac{N_{eff}}{N}$ always.

This paper deduces, and then relies on, a new sharp lower bound as indicated by the following proposition.

**Proposition 4.1.** *For $\nu \in \{1, N\}$,*

$$\inf_{\omega \in A_1} \varphi_1(\omega) = \varphi_1(\zeta_{[2]}) = \tfrac{1}{2},$$

$$\inf_{\omega \in A_N} \varphi_N(\omega) = \varphi_N(\zeta_{[N]}) = 1.$$

*Otherwise, if $2 \leq \nu \leq N - 1$, setting the point $p_\nu \in \tilde{\Delta}$ as*

$$p_\nu = \zeta_{[N]} + \frac{r_{\nu+1}}{r_1}(\zeta_{[1]} - \zeta_{[N]}) \in \overline{A_\nu},$$

*the following equality holds:*

$$\inf_{\omega \in A_\nu} \varphi_\nu(\omega) = \varphi_\nu(p_\nu) = \frac{\nu}{N} + \frac{N - \nu}{N} \sqrt{\frac{N - \nu - 1}{(\nu + 1)(N - 1)}}.$$

A proof of Proposition 4.1 covering the cases $\nu \geq 2$ is presented in Appendix A. Since we are interested in regimes where $N_{eff} = \rho N$, $N \gg 1$, the proof for the $\nu = 1$ case is omitted.

In terms of $s_{eff}$, together with the observations of Section 3.1, Proposition 4.1 yields the following theorem.

**Theorem 4.2.** *For all non-zero $s \in \mathbb{R}^N$ with $2 \leq N_{eff} < N$, one has the inequality*

$$1 - s_{eff} \leq \frac{N - N_{eff}}{N}\left(1 - \sqrt{\frac{N - N_{eff} - 1}{(N_{eff} + 1)(N - 1)}}\right)$$

$$\approx \frac{N - N_{eff}}{N}\left(1 - \sqrt{\frac{N - N_{eff}}{N N_{eff}}}\right). \tag{3}$$

### 4.2. Upper bound of the Performance drop by using EMP

This section demonstrates that a lower bound on the effective mass is essential for bounding the performance drop

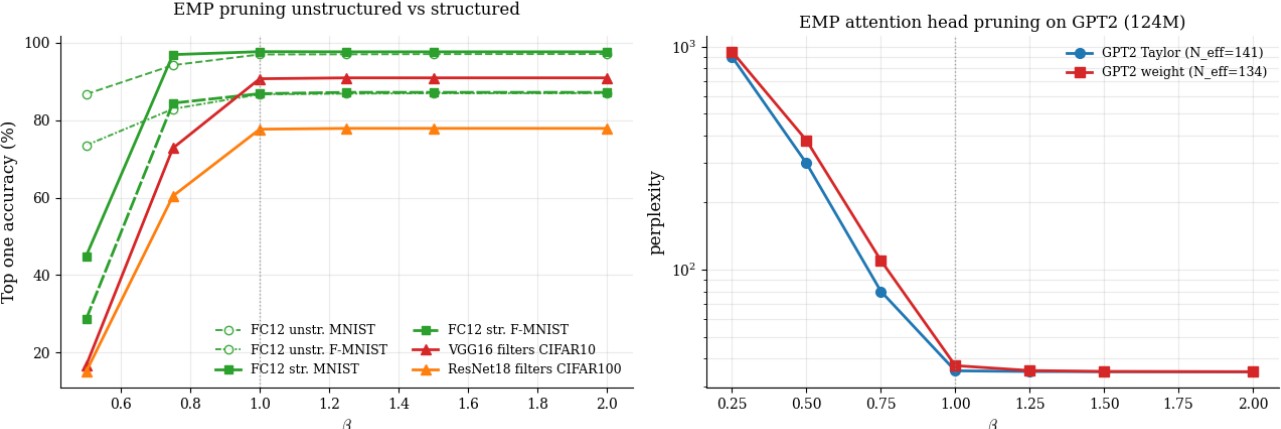

*Figure 3.* Test Accuracy and perplexity of EMP-pruned models across different values of $\beta$. We examine 6 discrete values of $\beta = \{0.5, 0.75, 1, 1.25, 1.5, 2\}$ to demonstrate that $N_{eff}$ is a robust pruning threshold across different models and methods. The left subplot demonstrates unstructured weight pruning (dashed; FC12 on MNIST and F-MNIST) with structured pruning (solid; FC12 neuron pruning, VGG16 filter pruning on CIFAR10, ResNet18 filter pruning on CIFAR100) The right subplot reports the WikiText 2 perplexity of GPT-2 (124M) under EMP based attention head pruning with the Taylor and weight norm importance criteria and evaluted from $\beta = 0.25$. For each importance criterion and across all tested architectures, $\beta = 1$ consistently marks the transition point where pruning beyond $N_{eff}$ leads to performance degradation, while pruning below $N_{eff}$ cuts efficiency. This uniformity in the statistics confirms that the effective number provides a principled, criterion-agnostic threshold.

in the transition from the dense well-trained network to a pruned network. In this section we study the case where the score vector $s$ coincides with the parameter vector, $\theta$ of the network (in other words, the parameters are scored according to their magnitude). Invoking Lemma 3.1 with $k = N_{eff}$, one has

$$\rho = \frac{N_{eff}}{N} \leq 1 - \frac{2\epsilon N}{\left\|\theta^* - \theta^k\right\|_2^2 \mathrm{Tr}(H) + 2\epsilon N}. \quad (4)$$

Rearranging equation 4 yields

$$\epsilon \leq \frac{1-\rho}{2N\rho} \mathrm{Tr}(H) \left\|\theta^* - \theta^{N_{eff}}\right\|_2^2,$$

where $\left\|\theta^* - \theta^{N_{eff}}\right\|_2^2$ can be bounded by

$$\left\|\theta^* - \theta^{N_{eff}}\right\|^2 \leq \left\|\|\theta\|_1 (\omega^* - \omega^k)\right\|^2$$
$$\leq \left\|\theta^*\right\|_1^2 \left\|(1 - s_{eff})\mathbf{1}_{[N - N_{eff}]}\right\|^2$$
$$= \left\|\theta^*\right\|_1^2 (1 - s_{eff})^2 (N - N_{eff}).$$

Hence, the asymptotic upper bound (as $N \to \infty$) of the loss change $\epsilon$ is

$$\epsilon \lesssim \left\|\theta^*\right\|_1^2 \mathrm{Tr}(H) \frac{(1-\rho)^4}{2\rho} \left(1 - \sqrt{\frac{1-\rho}{N\rho}}\right)^2. \quad (5)$$

The right panel of Figure 2 shows the relationship between $\epsilon/(\left\|\theta^*\right\|_1^2 \mathrm{Tr}(H))$ and $N_{eff}$. For $N = 1000$, the value of $\epsilon/(\left\|\theta^*\right\|_1^2 \mathrm{Tr}(H))$ is almost equal to 0 if $\rho > 0.2$.

---

**Algorithm 1** Effective Model Pruning

**Require:** Score vector $S \in \mathbb{R}^N$
**Ensure:** Binary mask $M \in \{0, 1\}^N$
1: $\omega \leftarrow |S|/\|S\|_1$ {normalize scores to a probability vector}
2: $N_{eff} \leftarrow \left\lfloor \left(\sum_{i=1}^N \omega_i^2\right)^{-1} \right\rfloor$ {effective number $N_{eff}$}
3: $N_{eff} \leftarrow \mathrm{clip}(N_{eff}, 1, N)$
4: $M \leftarrow \mathbf{0}_N$
5: $\pi \leftarrow \mathrm{argTopK}(|S|, N_{eff})$ {indices of the $N_{eff}$ largest entries in $|S|$}
6: **for** $i \in \pi$ **do**
7: $\quad M_i \leftarrow 1$
8: **end for**
9: **return** $M$

---

We test the performance of pruning Fully-Connected networks (FCs), AlexNet (Krizhevsky et al., 2012) and VGG16 (Simonyan & Zisserman, 2014) on CIFAR10 (Krizhevsky, 2009), ResNet18 and ResNet50 (He et al., 2016) on CIFAR100, and TinyImageNet (Le & Yang, 2015) with $N_{eff}$ threshold in Section 5.1. As shown in Table 1, test outcomes indicate that the loss change between the dense network and the corresponding EMP pruned network is almost 0 ($\epsilon \leq 0.1$).

Note that the upper bound on $\epsilon$ is derived under the weight-magnitude pruning criterion, and this guarantee does not

*Table 1.* Loss change between the dense models and the corresponding EMP pruned models.

| Dataset | Model | Dense Loss | Sparsity(%) | EMP Loss | $\epsilon$ |
|---------|-------|-----------|-------------|----------|------------|
| CIFAR10 | FC5 | 1.2582 | 47.41 | 1.2384 | 0.0198 |
| | FC12 | 1.5123 | 42.89 | 1.4454 | 0.0669 |
| | AlexNet | 0.4664 | 62.22 | 0.4286 | 0.0378 |
| | VGG16 | 0.4234 | 59.47 | 0.3184 | 0.1050 |
| CIFAR100 | ResNet18 | 0.8740 | 56.20 | 0.9287 | 0.0547 |
| | ResNet50 | 0.8586 | 54.74 | 0.8387 | 0.0199 |
| TinyImagenet | ResNet18 | 2.3028 | 53.37 | 2.2814 | 0.0214 |
| | ResNet50 | 2.0213 | 48.10 | 1.9853 | 0.0360 |

extend to alternative pruning strategies.[1] In contrast, the lower bound on the effective mass $s_{eff}$ can be generalized across different criteria, thereby providing potential upper bounds that quantify performance differences.

### 4.3. EMP Algorithm

We provide the pseudo-code of the proposed EMP approach in Algorithm 1. Given a score matrix $S$, the probabilistic vector $\omega$ is derived by normalizing the absolute value $|S|$ with its 1-norm. We then compute the effective number $N_{eff}$ and multiply with the coefficient $\beta$, which is an option to meet the specific sparse requirement in practical deployment[2]. The optional coefficient $\beta$ also helps to verify the robustness of $N_{eff}$ by range $\beta \in [0.5, 2.0]$. Since we multiply a potential larger than 1 coefficient, the effective number $N_{eff}$ needs to be constrained within the range of $[1, N]$. We then build a binary mask $M \in \{0, 1\}^N$. Set the $i$-th entry of $M$ to 1 for every $i \in \pi$. The algorithm has a time complexity of $O(N \log N)$.

Algorithm 1 normalizes the score vector via $\omega_i = |s_i|/|s|_1$, so only the magnitude of each entry determines its contribution. This is the natural convention for criteria in which magnitude itself encodes importance, such as weight magnitude, Taylor importance, and activation norms. For criteria in which the sign carries semantic meaning, namely those under which positive entries indicate retention and negative entries indicate removal, it would be necessary for the practitioner to apply a monotone map $\eta : \mathbb{R} \to \mathbb{R}_{\geq 0}$ to the entries of the score vector, prior to invoking EMP. For example, a shift (e.g., $s_i = s_i - \min_j s_j$), or the exponential map would render each entry nonnegative while preserving the ordering of the original scores together with the semantics

of the sign.

## 5. Experiments

In this section, we show that EMP can be applied to different network architectures, pruning criteria, and pruning objects. Through different values of the coefficient $\beta$, we demonstrate $N_{eff}$ is a robust effective pruning threshold across criterion and architectures. We examine the EMP method across several model types: FCs and CNNs in Section 5.1, Kolmogorov-Arnold Networks (KAN) in Section 5.2, and Large Language Models (LLMs) in Section 5.4. Featurewise pruning results are presented in Section 5.5.

### 5.1. FC models and CNNs

We first confirm that the $N_{eff}$ threshold yields negligible loss differences $\epsilon$ between the dense model and the EMP-pruned model. Specifically, we evaluate FC5, FC12, AlexNet, and VGG16 on CIFAR-10, as well as ResNet18 and ResNet50 on CIFAR-100 and TinyImageNet, using the $N_{eff}$ threshold applied to the magnitude pruning criterion.

Table 1 demonstrates that the loss difference between the dense network and the EMP pruned network remains negligible ($\epsilon \leq 0.105$) across all tested models. To examine the robustness of $N_{eff}$ as a pruning threshold across different model architectures, EMP is applied on FC2, FC5 and FC12, trained on MNIST and Fashion-MNIST. The detailed training and pruning settings are reported in Appendix B.1.

Figure 3 indicates $\beta = 1$ is the optimal setting across all tested models. For $\beta < 1$, the model will prune more entries than $N_{eff}$, causing model accuracy to consistently decline across all configurations. Conversely, for $\beta > 1$ accuracy plateaus, suggesting that any further increase would unnecessarily reduce model sparsity, yielding no significant performance gain. To fit the specific sparsity requirement for the hardware, $\beta$ can still serve as an optional adjustment. In addition to unstructured weight pruning, Figure 3 reports EMP under structured pruning, applied at the neuron level

---

[1]Nevertheless, one expects that, for a known and sufficiently smooth scoring function, bounds on first and second derivatives could be used for deriving principles analogous to the one reflected in equation 5.

[2]$\beta$ is not a hyperparameter requiring tuning. $\beta$ is introduced only as a deployment knob when a hardware constraint mandates a sparsity level lower than $N_{eff}/N$

*Table 2.* EMP under structured pruning at $\beta = 1$. The pruning component is the neuron for FC models and the filter for CNNs. The score vector is the magnitude of the structured unit.

| Dataset | Architecture | Dense Acc. (%) | EMP Acc. (%) | $\Delta$Acc. (pp) |
|---------|-------------|----------------|--------------|-------------------|
| MNIST | FC5 | 97.35 | 97.41 | $+0.06$ |
| MNIST | FC12 | 97.07 | 97.68 | $+0.61$ |
| F-MNIST | FC5 | 84.57 | 86.44 | $+1.87$ |
| F-MNIST | FC12 | 86.97 | 86.84 | $-0.13$ |
| CIFAR10 | VGG16 | 91.12 | 90.69 | $-0.43$ |
| CIFAR100 | ResNet18 | 77.88 | 77.67 | -0.21 |

for FC5 and FC12 and at the filter level for VGG16 and ResNet18 (numerical results in Table 2.)

## 5.2. KAN

From (Liu et al., 2024) Section 2.5.1, the incoming score for $i$-th node in $l$-th layer and the outgoing score are defined as

$$I_{l,i} = \max_k(\|\phi_{l-1,i,k}\|_1), \quad O_{l,i} = \max_j(\|\phi_{l+1,j,i}\|_1).$$

In Liu et al. (2024), each node in the network is pruned when both the incoming score and the outgoing score fall below a predefined threshold $\theta$. Instead of this predefined hyperparameter $\theta$, we combine EMP with the criterion $s = \min\{I_{l,i}, O_{l,i}\}$ and preserve the nodes with highest $N_{eff}$ scores entries per layer.

We performed numerical experiments using a KAN network with the initial width $[28 \times 28, 64, 10]$ on the MNIST dataset. After training for 10 epoch the validation loss reached 0.0923 with a validation accuracy of 97.15%. By applying EMP to KAN, the network structure changed to $[28 \times 28, 47, 10]$ with the validation loss increasing to 0.1810 and accuracy dropping to 94.36%.

## 5.3. Transformer and Attention

We apply EMP to attention head pruning in Transformers, following the observation that many attention heads are redundant and can be removed with limited effect on downstream performance (Michel et al., 2019).

**Experimental Setup.** We evaluate EMP-based head pruning on GPT-2 with 12 layers and 12 attention heads per layer, yielding 144 total heads. No fine-tuning is applied after pruning. We use WikiText-2 (Merity et al., 2016) for PPL evaluation. A calibration set of 50,000 tokens from the training split is used to compute head importance scores.

**Head Importance Criteria.** We consider two importance criteria for scoring attention heads. The first is Taylor importance, which computes the first-order Taylor approximation of each head's contribution to the loss. For head $h$ at layer

*Table 3.* EMP attention head pruning on GPT-2 evaluated on WikiText-2. Sparsity denotes the fraction of pruned heads.

| Criterion | $N_{eff}$ | Sparsity (%) | PPL |
|-----------|-----------|--------------|-----|
| Taylor | 141.4 | 2.1 | 35.20 |
| Weight | 134.0 | 6.9 | 37.14 |

$\ell$, the Taylor importance is defined as

$$I_h^\ell = \left| \frac{\partial \mathcal{L}}{\partial Y^{\ell,h}} \odot Y^{\ell,h} \right|,$$

where $Y^{\ell,h} \in \mathbb{R}^{T \times d_h}$ denotes the output of head $h$ at layer $\ell$, and the absolute value and Hadamard product are taken elementwise before summation. This criterion requires a forward and backward pass over the calibration set.

The second criterion is weight norm, a data-free measure that scores each head by the Frobenius norm of its output projection weights as

$$I_h^\ell = \|W_O^h\|_F,$$

where $W_O^h$ is the output projection matrix for head $h$ and $\|\cdot\|$ denotes the Frobenius norm. This criterion requires no calibration data and can be computed directly from the pretrained weights.

**Results.** Table 3 summarizes the EMP head pruning results on GPT-2. The baseline PPL on the WikiText-2 test set is 34.86. The Taylor importance criterion yields $N_{eff} = 141.4$, indicating that importance is spread almost uniformly across all 144 heads. Correspondingly, EMP retains 141 heads at $\beta = 1$ with only a 1.0% PPL increase (from 34.86 to 35.20). The weight norm criterion produces $N_{eff} = 134.0$, and EMP prunes 10 heads with a 6.5% PPL increase (from 34.86 to 37.14).

When $\beta < 1$, pruning becomes more aggressive and performance degrades dramatically. These results confirm that $N_{eff}$ identifies the boundary between effective and redundant heads.

**Robustness Across $\beta$.** As shown in Figure 3 (right y-axis), the PPL remains stable for $\beta \geq 1$ and increases sharply for $\beta < 1$. This pattern mirrors the accuracy curves observed

*Table 4.* $N_{eff}$ is an adaptive threshold for different criterion with different models. The goal is to determine the optimal sparsity level for a given scoring method, such that the resulting change in performance remains small. By using the threshold $N_{eff}$, different scoring methods yield different optimal sparsity levels, which can serve as a criterion for assessing the quality of a method. For example, weight magnitude is known to perform poorly for LLM pruning at high sparsity. However, EMP magnitude indicates that, when the sparsity is reduced to 36.63%, magnitude based pruning can preserve performance close to that of the dense model and comparable to the Wanda method.

| Method | Avg Sparsity(%) | Avg $\Delta$PPL | Avg $\Delta$Acc.(%) |
|---|---|---|---|
| Wanda | 50.00 | +0.799 | -1.40 |
| Magnitude | 50.00 | +2.982 | -2.60 |
| EMP-Wanda | 40.47 | +0.678 | -1.37 |
| EMP-Magnitude | 36.63 | +0.752 | -0.93 |

for weight pruning in MLPs, demonstrating that $N_{eff}$ serves as a universal pruning threshold. The consistency of this transition point across different architectures, datasets, and pruning criteria validates the theoretical foundation of EMP.

**Connection to Gated Attention.** Gated attention mechanisms augment the standard attention operator with learnable gates that modulate information flow (Qiu et al., 2025). EMP can be viewed as a deterministic, hard-gating variant where the gate is binary and determined entirely by the effective number rather than learned parameters. Unlike learned gating, EMP requires no additional parameters or training and provides a theoretically grounded threshold with the guaranteed lower bound on preserved importance mass established in Theorem 4.2.

This connection suggests a potential hybrid approach: using EMP as an initialization or regularization target for learned gating mechanisms, or applying EMP at inference time to dynamically adjust the number of active heads based on input complexity. We leave systematic exploration of these directions for future work.

### 5.4. LLMs

In this subsection, following the experimental setup in (Sun et al., 2023), we evaluate LLama (Touvron et al., 2023a) and LLama-2 (Touvron et al., 2023b) model families' PPL (PPL) on Wikitext and zero-shot accuracy across 7 sub-tasks. We examine the models under the pruning criterion magnitude, Wanda and the corresponding criterion composed with EMP: EMP-magnitude and EMP-Wanda. We show the average sparsity for EMP methods, the average PPL change and the average accuracy change for all 7 models in Table 4. The detail sparsity and PPL for each model and the detail accuracy of each task is shown in Appendix B.

From Table 4, the EMP based methods are able to perform a comparable performance with the dense network. Notably,

the EMP-magnitude method reduced the PPL and increased the accuracy compared with the fixed sparsity magnitude method, in the cost of lower sparsity.

Beyond this aggregate comparison, Table 4 also indicates that EMP can effectively "rehabilitate" plain magnitude pruning in the LLM regime. While fixed 50% magnitude pruning exhibits the largest degradation among the baselines (average $\Delta PPL \approx +3.0$ and $\Delta Acc \approx -2.6\%$), switching to the EMP-selected sparsity level reduces the average sparsity to about 37% yet brings the PPL and accuracy back to within roughly 1 PPL and 1% accuracy of the dense models. In other words, the main failure mode of magnitude pruning at 50% sparsity appears to be the rigid global budget, not the score itself: once the sparsity is chosen adaptively via EMP, even this simple criterion becomes competitive with the dense network at a substantially lower parameter count.

### 5.5. Featurewise Effective Pruning

In this section, we show that EMP can also be applied to image features. We apply EMP to an RGB image by processing each RGB channel independently. For a given channel, we defined the score matrix $s \triangleq X_c - \mu_c$, where $X_c$ denotes the pixel values of channel $c \in \{R, G, B\}$, and $\mu_c$ is the corresponding channel mean. Two EMP-based pruning strategies were considered: EMP Global Magnitude and EMP Patch Magnitude. EMP Global Magnitude uses the score matrix $s$ over the entire channel, and pruning was performed at the global scale, while EMP Patch Magnitude partitions the image into $4 \times 4$ non-overlapping patches, and EMP pruning was applied independently within each patch. After pruning, we restored the mean by adding $\mu_c$ back to each channel, followed by concatenation of the R, G, and B channels to reconstruct the pruned image.

To quantify the quality of the pruned image, we measured sparsity, structural similarity index (SSIM), and peak signal-to-noise ratio (PSNR) between the original and pruned images. The results are summarized in Fig 4. These results indicate that EMP at the feature level can aggressively zero out low-importance coordinates while preserving nearly all task-relevant signals in the remaining ones. In particular, the high PSNR/SSIM at nontrivial sparsity suggests that the effective mass of the retained features remains close to one, so that most of the normalized energy of the representation is carried by a relatively small subset of dimensions. In a learning setting, this motivates using EMP as an adaptive feature-selection mechanism applied to intermediate activations: by activating only the top-$N_{eff}$ features per layer, one can potentially reduce the number of units participating in forward and backward passes, thereby lowering computational cost and gradient noise, while maintaining the expressive power and predictive performance of the original dense representation. A systematic study of such training-

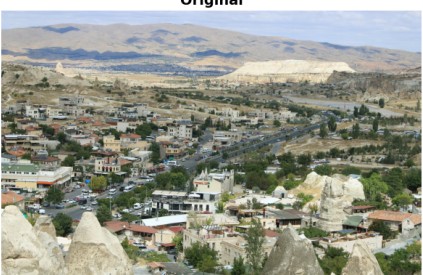 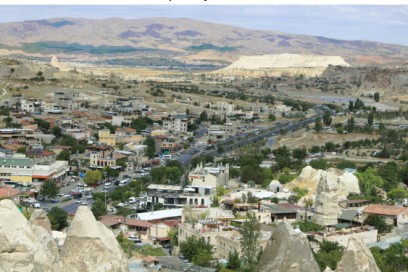 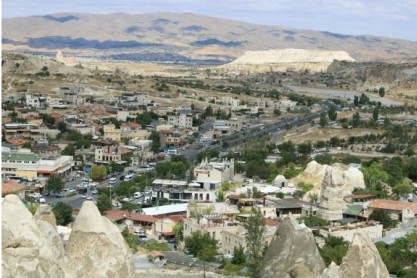

*Figure 4.* EMP magnitude pruning on an RGB image. Left: Original image Middle: EMP global magnitude pruning applied independently to each RGB channel. Right: patchwise EMP magnitude pruning with local EMP applied on non-overlapping $4 \times 4$ patches. The global method retains $PSNR = 29.4dB$ and $SSIM = 0.912$ at sparsity 26.7%, while the patchwise method achieves higher fidelity ($PSNR = 38.3dB, SSIM = 0.991$) at increased sparsity 32.3%.

time featurewise EMP, including its impact on convergence speed and generalization, is an interesting direction for future work.

## 6. Conclusion

In this paper, we introduced a universal pruning threshold $N_{eff}$, which is agnostic to the scoring criterion, the network architecture, and the pruning paradigm. Since EMP rests only on statistical properties of the score $s$, it may be paired with a variety of pruning criteria, even feature-level image pixels. Experiments demonstrate that, on the same model, EMP suggests different optimal sparsity levels for different pruning methods, where optimal sparsity is understood as the threshold above which the loss change is negligible in the transition from the dense network to the pruned network. This phenomenon extends across a wide variety of network architectures, suggesting that the natural sparsity measure in relation to performance loss under pruning is not $\rho = \frac{N_{ret}}{N}$, but $\beta = \frac{N_{ret}}{N_{eff}}$, as $N_{eff}$ appears to be a threshold for significant performance loss, see Figure 3. In other words, EMP provides a measure of quality of a pruning method (for a given model), the number $\frac{N}{N_{eff}}$ (thus $\rho$ equals $\beta$ divided by the quality) while $\beta$ serves as a "sparsity knob" that can be tuned to meet specific sparsity requirements. A systematic study of gated EMP, featurewise EMP and its impact on convergence speed, generalization, attention sinks and large-scale LLM finetuning is a natural direction for future work. Furthermore, future development may also include employing EMP during training in an adaptive fashion, pruning poorly performing features periodically, as they emerge.

## Acknowledgments

The authors thank the reviewers for their careful reading of the manuscript and for their constructive suggestions, which substantially improved the clarity and rigor of our presentation.

This research was supported in part by Air Force Office of Scientific Research award number FA9550-22-1-0429, FA9550-19-1-0169. Any opinions, findings and conclusions or recommendations expressed in this material are those of the author(s) and do not necessarily reflect the views of the sponsoring agency.

## Impact Statement

This work introduces Effective Model Pruning (EMP), a general selection rule derived from a distributional statistic that quantifies concentration through an effective number criterion. Because EMP depends only on the empirical distribution of a nonnegative importance score, it is model agnostic and task agnostic, and it can be applied wherever one can compute scores over a set of candidates. In particular, EMP does not require access to second order information, does not assume a specific architecture, and does not rely on a training objective beyond the availability of calibration data for estimating score statistics.

The primary positive impact of EMP is to provide a unified and principled mechanism for adaptive selection and sparsification. The same criterion can be used for model pruning at the level of weights, channels, neurons, or attention edges, for feature extraction by selecting informative dimensions, and as a post hoc sparsification layer that induces hard gates without introducing additional learnable parameters. When used in attention, EMP operates on pre softmax scores and therefore can be interpreted as hard gated attention, which can reduce pathological concentration phenomena such as attention sinks by suppressing low importance edges before normalization.

EMP also has potential utility beyond pruning. Since effective number measures the intrinsic diversity of a score distribution, it can serve as a diagnostic for representation collapse, as an adaptive rule for retaining components in low rank approximations, and as a score driven alternative to choosing dimensionality in procedures related to principal component selection. These uses may help practitioners avoid manual threshold tuning by replacing fixed sparsity ratios with a data adaptive selection rule.

Potential negative impacts arise from the fact that reduce redundancy that sometimes hurts the generality, and it may increase susceptibility to adversarial perturbations in some regimes.

Overall, EMP provides a broadly applicable statistical tool for adaptive selection that can be instantiated as feature selection, component retention, model pruning, sparsification, and hard gating, while remaining simple to implement and compatible with existing training and inference pipelines.

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

# A. Proof of Proposition 4.1

*Proof.* For $\nu = N$, the result is trivial because $A_\nu$ is a single point. Now assume $1 < \nu < N$. Let $\omega \in \tilde{\Delta}$ be fixed. Clearly, $p_\nu$ minimizes $\varphi_\nu$ along the interval $[\zeta_{[1]}, \zeta_{[N]}]$ (note that this interval is the longest edge of $\tilde{\Delta}$). Therefore, without loss of generality, we may assume $\omega \notin [\zeta_{[1]}, \zeta_{[N]}]$. Then, there exist indices $1 < i \leq \nu$ and $\nu < j \leq N$ such that $\omega_i > \omega_j$. Among the pairs $(i, j)$ satisfying this condition, select the one with smallest $j - i$ and smallest $j$. We denote this pair by $i = i(\omega)$ and $j = j(\omega)$. Let

$$H_\omega = \{x \in \mathbb{R}^N : \left\langle \zeta_{[N]} \mid x \right\rangle = 0, \langle \omega \mid x \rangle \geq 0\}.$$

It follows that $H_\omega$ is contained in the tangent space to $\tilde{\Delta}$, as well as

$$\|\omega + \epsilon x\| \geq \|\omega\|$$

for every $\epsilon > 0$.

We construct a vector $x \in H_\omega$ as

$$x_1 := 1, \; x_i := t, \; x_j := -(1 + t), \; \text{and } x_k := 0 \text{ for } k \neq 1, i, j,$$

where

$$t = \frac{\omega_j - \omega_1}{\omega_i - \omega_j} \leq -1,$$

because $\omega_1 \geq \omega_i > \omega_j \geq \omega_N$. Also, note that $x$ is orthogonal to $\zeta_{[N]}$, implying that it is in the tangent space to the simplex $\Delta$. Furthermore, by the choice of the indices $i, j$, there exists $\epsilon > 0$ small enough that the ordering among the weights is preserved upon adding $\epsilon x$ to $\omega$, and therefore $\omega + \epsilon x \in \tilde{\Delta}$.

Consider the derivative of $\varphi_\nu$ at $\omega$ in the direction of $x$:

$$D\varphi_\nu\big|_\omega x = \nu \zeta_{[\nu]}^\top x = (1 + t) \leq 0.$$

The function $\varphi_\nu$ is linear. Therefore, setting $\epsilon_1 := \max\{\epsilon > 0 : \omega + \epsilon x \in \tilde{\Delta}\}$ gives rise to a point $\omega^1 := \omega + \epsilon_1 x$ with $\varphi_\nu(\omega^1) < \varphi_\nu(\omega)$ and with the property that either $\omega_1 \in [\zeta_{[1]}, \zeta_{[N]}]$ or $j(\omega^1) - i(\omega^1) < j(\omega) - i(\omega)$. Therefore, the process of augmenting $\omega$ into $\omega_1$ may be repeated at most $j - i$ times, generating a sequence of vectors $\omega^1, \ldots, \omega^k$ with $k \leq j - i$ and $\omega^k \in [\zeta_{[1]}, \zeta_{[N]}]$, and such that $\varphi_\nu(\omega) > \varphi_\nu(\omega^1) > \ldots > \varphi_\nu(\omega^k)$. Since $\omega^k \in [\zeta_{[1]}, \zeta_{[N]}]$, we finally have $\varphi_\nu(\omega^k) \geq \varphi_\nu(p_\nu)$ (with equality only if $\omega \neq p_\nu$ in the first place), as $\|\omega_k\| \geq \|p_\nu\|$.

Given there exist such a point $p_\nu$, let the vector $u$ represent the direction from $\zeta_N$ point to $\zeta_{[1]}$ as

$$u = \left[\frac{N-1}{N}, -\frac{1}{N}, \ldots, -\frac{1}{N}\right].$$

Then the coordinate of $p_\nu$ is

$$p_\nu = c + r_{\nu+1} \frac{u}{\|u\|}$$

$$= \begin{bmatrix} \frac{1}{N} + \frac{1}{N}\sqrt{\frac{(N-1)(N-\nu-1)}{\nu+1}} \\ \frac{1}{N}\left(1 - \sqrt{\frac{N-\nu-1}{(N-1)(\nu+1)}}\right) \\ \cdots \\ \frac{1}{N}\left(1 - \sqrt{\frac{N-\nu-1}{(N-1)(\nu+1)}}\right) \end{bmatrix}^\top$$

and the one-norm of the projection $T_\nu(p_\nu)$ is

$$\varphi_\nu(\omega) = \|T_\nu(p_\nu)\|_1 = \frac{\nu}{N} + \frac{N-\nu}{N}\sqrt{\frac{N-\nu-1}{(\nu+1)(N-1)}}.$$

The proof is now complete. $\qquad\square$

*Table 5.* Network structure of FC models

| Model | Layer Width |
|-------|-------------|
| FC2   | $100, 10$ |
| FC5   | $1000, 600, 300, 100, 10$ |
| FC12  | $1000, 900, 800, 750, 700, 650, 600, 500, 400, 200, 100, 10$ |

## B. Experiment Details

In this section, we provide the experimental details corresponding to Section 5. To further demonstrate that EMP is a context-agnostic pruning method, we additionally evaluate loss change and gradient saliency pruning method on VGG16.

### B.1. Training details

This section specifies the training configures of all the dense models used in this paper. We report dataset, architecture, optimizer, schedule, and pruning settings precisely.

**Datasets and preprocessing.** We use the standard training and test splits of MNIST, FashionMNIST, CIFAR-10, CIFAR-100, and TinyImageNet. Images are resized to the model's declared input size: 28 for MNIST/FashionMNIST, 32 for most CIFAR models, 224 for AlexNet (by upsampling), and 64 for TinyImageNet.

**Architectures.** Fully connected baselines are FC2, FC5, and FC12 (Table 5). For CNNs, we use AlexNet and VGG16 on CIFAR-10. For CIFAR-100 and Tiny ImageNet we use ResNet-18/50.

**Optimization.** Unless stated, batch size is 128 and epochs are 200. Optimizers and hyperparameters follow the configuration dictionary:

- **MNIST/FashionMNIST (FC2/5/12):** Adam, learning rate $10^{-4}$, no cosine schedule, no warmup, weight decay 0; FC2/5 trained for 5 epochs and FC12 for 10 epochs.

- **CIFAR-10:** FC5/FC12 with SGD, learning rate 0.01, no cosine schedule, no warmup, weight decay 0; VGG16 and AlexNet with SGD, learning rate 0.01, cosine schedule enabled, 5 warmup epochs, weight decay $5 \times 10^{-4}$.

- **CIFAR-100:** ResNet-18/50 with SGD, learning rate 0.1, cosine schedule, 5 warmup epochs, weight decay $5 \times 10^{-4}$.

- **Tiny ImageNet:** ResNet-18/50 with SGD, learning rate 0.01, cosine schedule, 5 warmup epochs, weight decay $5 \times 10^{-4}$.

SGD uses momentum 0.9. Adam uses its standard defaults unless otherwise noted.

**Learning-rate schedule.** When cosine is enabled with warmup $W$, the step-$t$ learning rate for total $T$ epochs is

$$\mathrm{lr}(t) = \begin{cases} \mathrm{lr}_0 \cdot \frac{t}{W}, & 0 \le t < W, \\ \mathrm{lr}_0 \cdot \frac{1}{2}\left(1 + \cos\left(\pi \frac{t-W}{T-W}\right)\right), & W \le t \le T. \end{cases}$$

Table 6 reports the detailed pruning results of EMP with $\beta = 1$, along with comparisons to their corresponding dense models.

### B.2. LLM

Meta LLama and LLama-2 checkpoints are from HuggingFace Hub. We load the pretrained, untuned weights and tokenizer; no fine-tuning or gradient updates occur. We run inference in $bfloat16$ with device map set to auto across 4× NVIDIA $B200$ GPUs.

*Table 6.* Pruning results of FC models at $\beta = 1$ on MNIST and Fashion-MNIST.

| Dataset | Model | Dense Acc. (%) | Prune type | Acc. (%) | Sparsity (%) | $\Delta$ Acc. (%) |
|---------|-------|----------------|------------|----------|--------------|-------------------|
| MNIST   | fc2   | 93.89          | block      | 93.39    | 34.13        | -0.50             |
|         |       | 93.89          | global     | 93.78    | 37.19        | -0.11             |
|         | fc5   | 97.35          | block      | 97.31    | 28.69        | -0.04             |
|         |       | 97.35          | global     | 97.31    | 29.56        | -0.04             |
|         | fc12  | 97.07          | block      | 96.95    | 27.71        | -0.12             |
|         |       | 97.07          | global     | 96.96    | 28.32        | -0.11             |
| F-MNIST | fc2   | 83.93          | block      | 83.17    | 33.86        | -0.76             |
|         |       | 83.93          | global     | 83.71    | 36.72        | -0.22             |
|         | fc5   | 84.57          | block      | 84.65    | 28.90        | +0.08             |
|         |       | 84.57          | global     | 84.84    | 29.63        | +0.27             |
|         | fc12  | 86.97          | block      | 86.87    | 27.58        | -0.10             |
|         |       | 86.97          | global     | 86.68    | 28.16        | -0.29             |

The detail of the sparsity, PPL, and mean accuracy across 7 subtasks are reported in Table 7. The detail accuracy for different tasks are reported in Table 8 for the LLama model family and Table 9 for the LLama-2 family models.

### B.3. additional EMP experiments

We evaluate the effectiveness of the $N_{eff}$ threshold on a VGG16 model trained on CIFAR-10. Three pruning criteria are considered: magnitude pruning, loss-change (Taylor) pruning, and gradient saliency pruning. We report accuracy, achieved sparsity, and FLOPs after pruning in Table 10.

For magnitude pruning $N_{eff}$ threshold yields a high sparsity regime with minimal accuracy drop, while for sensitivity-based criteria it trades a very low sparsity with the model performance.

*Table 7.* Detail PPL and 7-task mean accuracy for LLama and LLama-2 families pruning model by Wanda, magnitude, EMP-Wanda, and EMP-magnitude.

| Model | Method | Sparsity (%) | PPL | ΔPPL | Mean Acc. (%) | ΔAcc (pp) |
|---|---|---|---|---|---|---|
| LLaMA 7B | Dense | 0.00 | 5.679 | +0.000 | 51.10 | +0.00 |
| | Wanda | 50.00 | 6.644 | +0.965 | 49.48 | -1.62 |
| | Magnitude | 50.00 | 11.002 | +5.323 | 48.42 | -2.68 |
| | EMP-Wanda | 40.60 | 6.362 | +0.683 | 50.34 | -0.76 |
| | EMP-Magnitude | 36.66 | 6.904 | +1.225 | 51.11 | +0.01 |
| LLaMA 13B | Dense | 0.00 | 5.090 | +0.000 | 53.60 | +0.00 |
| | Wanda | 50.00 | 5.836 | +0.746 | 52.00 | -1.60 |
| | Magnitude | 50.00 | 11.587 | +6.497 | 49.25 | -4.35 |
| | EMP-Wanda | 40.68 | 5.907 | +0.817 | 52.74 | -0.86 |
| | EMP-Magnitude | 36.58 | 6.666 | +1.576 | 52.02 | -1.58 |
| LLaMA 30B | Dense | 0.00 | 4.101 | +0.000 | 54.84 | +0.00 |
| | Wanda | 50.00 | 4.890 | +0.789 | 54.16 | -0.68 |
| | Magnitude | 50.00 | 5.553 | +1.452 | 53.57 | -1.27 |
| | EMP-Wanda | 40.18 | 4.687 | +0.586 | 52.90 | -1.94 |
| | EMP-Magnitude | 36.60 | 4.511 | +0.410 | 54.82 | -0.02 |
| LLaMA 65B | Dense | 0.00 | 3.531 | +0.000 | 59.28 | +0.00 |
| | Wanda | 50.00 | 4.267 | +0.736 | 57.40 | -1.88 |
| | Magnitude | 50.00 | 4.724 | +1.193 | 55.83 | -3.45 |
| | EMP-Wanda | 39.99 | 4.060 | +0.529 | 57.08 | -2.20 |
| | EMP-Magnitude | 36.61 | 3.865 | +0.334 | 56.95 | -2.33 |
| LLaMA-2 7B | Dense | 0.00 | 5.470 | +0.000 | 51.59 | +0.00 |
| | Wanda | 50.00 | 6.410 | +0.940 | 50.27 | -1.32 |
| | Magnitude | 50.00 | 9.712 | +4.242 | 48.52 | -3.07 |
| | EMP-Wanda | 41.07 | 6.513 | +1.043 | 49.78 | -1.81 |
| | EMP-Magnitude | 36.70 | 6.561 | +1.091 | 51.16 | -0.43 |
| LLaMA-2 13B | Dense | 0.00 | 4.881 | +0.000 | 53.64 | +0.00 |
| | Wanda | 50.00 | 5.591 | +0.710 | 52.12 | -1.52 |
| | Magnitude | 50.00 | 5.850 | +0.969 | 52.59 | -1.05 |
| | EMP-Wanda | 40.48 | 5.468 | +0.587 | 51.96 | -1.68 |
| | EMP-Magnitude | 36.62 | 5.162 | +0.281 | 52.93 | -0.71 |
| LLaMA-2 70B | Dense | 0.00 | 3.319 | +0.000 | 60.00 | +0.00 |
| | Wanda | 50.00 | 4.026 | +0.707 | 58.81 | -1.19 |
| | Magnitude | 50.00 | 4.514 | +1.195 | 57.70 | -2.30 |
| | EMP-Wanda | 40.32 | 3.821 | +0.502 | 58.74 | -1.26 |
| | EMP-Magnitude | 36.66 | 3.662 | +0.343 | 58.57 | -1.43 |

*Table 8.* Detailed zero-shot accuracy across 7 tasks (LLaMA).

| Model | Method | BoolQ | RTE | HellaSwag | WinoGrande | ARC-e | ARC-c | OBQA | Mean |
|---|---|---|---|---|---|---|---|---|---|
| LLaMA 7B | Dense | 69.63 | 53.07 | 54.25 | 49.17 | 67.05 | 38.74 | 25.80 | 51.10 |
| | Wanda 50% | 70.24 | 52.71 | 51.12 | 50.67 | 62.25 | 35.75 | 23.60 | 49.48 |
| | Magnitude 50% | 67.25 | 52.71 | 48.95 | 50.04 | 60.73 | 34.64 | 24.60 | 48.42 |
| | EMP-Wanda 40.6% | 72.05 | 53.07 | 52.59 | 48.62 | 64.86 | 37.63 | 23.60 | 50.34 |
| | EMP-Magnitude 36.66% | 72.81 | 53.43 | 52.94 | 49.41 | 65.32 | 37.46 | 26.40 | 51.11 |
| LLaMA 13B | Dense | 66.36 | 54.51 | 57.46 | 48.07 | 74.33 | 43.86 | 30.60 | 53.60 |
| | Wanda 50% | 68.32 | 53.07 | 54.87 | 47.83 | 70.24 | 40.87 | 28.80 | 52.00 |
| | Magnitude 50% | 65.57 | 51.99 | 49.47 | 49.64 | 62.54 | 36.52 | 29.00 | 49.25 |
| | EMP-Wanda 40.68% | 68.20 | 57.04 | 55.93 | 47.75 | 69.28 | 41.38 | 29.60 | 52.74 |
| | EMP-Magnitude 36.58% | 65.84 | 53.07 | 54.95 | 47.99 | 70.37 | 42.32 | 29.60 | 52.02 |
| LLaMA 30B | Dense | 66.91 | 53.79 | 60.75 | 49.64 | 75.08 | 46.93 | 30.80 | 54.84 |
| | Wanda 50% | 68.99 | 54.51 | 58.89 | 49.96 | 72.85 | 44.28 | 29.60 | 54.16 |
| | Magnitude 50% | 70.46 | 52.35 | 56.40 | 50.20 | 71.93 | 43.43 | 30.20 | 53.57 |
| | EMP-Wanda 40.18% | 67.25 | 52.35 | 59.31 | 50.36 | 71.00 | 42.66 | 27.40 | 52.90 |
| | EMP-Magnitude 36.6% | 67.68 | 53.43 | 59.85 | 50.28 | 74.58 | 45.90 | 32.00 | 54.82 |
| LLaMA 65B | Dense | 80.31 | 66.79 | 62.32 | 50.20 | 74.87 | 46.67 | 33.80 | 59.28 |
| | Wanda 50% | 80.15 | 60.29 | 60.49 | 50.43 | 74.16 | 45.31 | 31.00 | 57.40 |
| | Magnitude 50% | 81.31 | 52.71 | 59.89 | 50.91 | 71.93 | 44.28 | 29.80 | 55.83 |
| | EMP-Wanda 39.99% | 79.88 | 61.73 | 60.46 | 50.20 | 72.39 | 45.90 | 29.00 | 57.08 |
| | EMP-Magnitude 36.61% | 81.04 | 54.51 | 61.81 | 50.36 | 73.70 | 46.25 | 31.00 | 56.95 |

*Table 9.* Detailed zero-shot accuracy across 7 tasks (LLaMA-2).

| Model | Method | BoolQ | RTE | HellaSwag | WinoGrande | ARC-e | ARC-c | OBQA | Mean |
|---|---|---|---|---|---|---|---|---|---|
| LLaMA-2 7B | Dense | 66.57 | 52.71 | 54.56 | 50.43 | 69.23 | 39.85 | 27.80 | 51.59 |
| | Wanda 50% | 72.39 | 52.71 | 51.12 | 49.57 | 65.74 | 36.18 | 24.20 | 50.27 |
| | Magnitude 50% | 64.86 | 53.79 | 50.49 | 49.49 | 62.79 | 34.04 | 24.20 | 48.52 |
| | EMP-Wanda 41.07% | 68.72 | 53.07 | 51.32 | 49.80 | 65.66 | 37.12 | 22.80 | 49.78 |
| | EMP-Magnitude 36.7% | 69.79 | 53.07 | 54.35 | 48.86 | 68.98 | 38.65 | 24.40 | 51.16 |
| LLaMA-2 13B | Dense | 66.82 | 52.71 | 57.54 | 48.54 | 73.32 | 45.39 | 31.20 | 53.64 |
| | Wanda 50% | 66.06 | 52.71 | 55.13 | 49.57 | 70.79 | 41.38 | 29.20 | 52.12 |
| | Magnitude 50% | 67.31 | 52.71 | 55.92 | 50.20 | 70.29 | 41.13 | 30.60 | 52.59 |
| | EMP-Wanda 40.48% | 63.55 | 52.71 | 56.05 | 49.49 | 70.41 | 42.49 | 29.00 | 51.96 |
| | EMP-Magnitude 36.62% | 65.60 | 52.71 | 57.57 | 49.09 | 72.31 | 42.83 | 30.40 | 52.93 |
| LLaMA-2 70B | Dense | 77.55 | 64.98 | 62.81 | 51.30 | 77.44 | 50.34 | 35.60 | 60.00 |
| | Wanda 50% | 81.38 | 60.65 | 61.14 | 50.99 | 75.63 | 48.29 | 33.60 | 58.81 |
| | Magnitude 50% | 82.45 | 57.76 | 60.15 | 52.09 | 73.36 | 45.90 | 32.20 | 57.70 |
| | EMP-Wanda 40.32% | 75.81 | 65.70 | 61.91 | 50.43 | 74.75 | 48.21 | 34.40 | 58.74 |
| | EMP-Magnitude 36.66% | 79.42 | 56.32 | 62.17 | 51.30 | 77.23 | 48.55 | 35.00 | 58.57 |

*Table 10.* Comparison of pruning criteria on VGG16 (CIFAR-10). Dense baseline: 91.12% accuracy, 626M FLOPs. Neff threshold uses $\beta = 1.0$.

| Method | Acc.(%) | Sparsity(%) | FLOPs |
|---|---|---|---|
| Dense | 91.12 | 0.0 | 626M |
| EMP-Magnitude | 90.98 | 59.5 | 397M |
| EMP-Loss change | 91.12 | 2.1 | 623M |
| EMP-Saliency | 90.97 | 25.7 | 579M |

*Table 11.* Pruning results on VGG16 trained on CIFAR-10. EMP pruning using $\beta \in \{0.8, 1.0, 1.2\}$.

| Criterion | Pruning Scheme | Acc.(%) | Sparsity (%) | FLOPs |
|---|---|---|---|---|
| Magnitude | Original (50%) | 90.99 | 50.0 | 436.7M |
| | Original (70%) | 90.64 | 70.0 | 348.7M |
| | Original (90%) | 69.08 | 90.0 | 221.0M |
| | Original (95%) | 10.00 | 95.0 | 157.9M |
| | EMP ($\beta = 0.8$) | 90.80 | 67.6 | 360.4M |
| | EMP ($\beta = 1.0$) | 90.98 | 59.5 | 396.9M |
| | EMP ($\beta = 1.2$) | 91.09 | 51.4 | 431.1M |
| Loss Change (Taylor) | Original (50%) | 88.14 | 50.0 | 490.8M |
| | Original (70%) | 49.86 | 70.0 | 410.1M |
| | Original (90%) | 10.00 | 90.0 | 261.0M |
| | Original (95%) | 10.00 | 95.0 | 169.8M |
| | EMP ($\beta = 0.8$) | 91.12 | 2.1 | 623.1M |
| | EMP ($\beta = 1.0$) | 91.12 | 2.1 | 623.1M |
| | EMP ($\beta = 1.2$) | 91.14 | 2.3 | 622.8M |
| Gradient Saliency | Original (50%) | 88.08 | 50.0 | 514.1M |
| | Original (70%) | 45.19 | 70.0 | 449.6M |
| | Original (90%) | 10.00 | 90.0 | 306.4M |
| | Original (95%) | 10.00 | 95.0 | 189.9M |
| | EMP ($\beta = 0.8$) | 91.10 | 20.6 | 590.7M |
| | EMP ($\beta = 1.0$) | 90.97 | 25.7 | 578.8M |
| | EMP ($\beta = 1.2$) | 90.82 | 30.9 | 566.2M |

