# OpenReview forum: "Effective Model Pruning : Measuring the Redundancy of Model Components"
_ICML.cc/2026/Conference — ICML 2026 spotlight_

### Official Review · Reviewer_zoFh · 2026-03-09

**Soundness:** 3
**Presentation:** 3
**Significance:** 3
**Originality:** 3
**Overall Recommendation:** 5
**Confidence:** 3

**Summary:**

While pruning is a widely used compression technique, solutions remain unsatisfactory, given that the definition of pruning budgets and schedules is mostly manual and requires careful tuning. The authors propose Effective Model Pruning (EMP), a score-distribution-driven rule for automatically selecting a pruning budget. They analyze the geometry of the $N_{eff}$ threshold to derive a lower bound on the preserved mass when retaining the top-$N_{\mathrm{eff}}$ entries. The claims are empirically evaluated on a broad range of architectures and tasks, including CNN/FC weight pruning, KAN node pruning, transformer head pruning (GPT-2), LLM weight pruning (LLaMA/LLaMA-2), and featurewise image pruning, demonstrating that EMP consistently identifies transition points.

**Compliance With Llm Reviewing Policy:**

Affirmed.

**Final Justification:**

I maintain my initial assessment.

**Key Questions For Authors:**

1. Please compare EMP to budget rules, even if naive. For example, percentile thresholds, or knee detection on performance–sparsity curves.
2. Please report the wall-clock/flops reduction, especially for the settings that employ structured pruning.

**Limitations:**

Yes

**Strengths And Weaknesses:**

**Strengths**
1. The authors present a technically simple and theoretically founded method.
2. Important theoretical contribution that addresses a general shortcoming of pruning methods.
3. I like the coherent storytelling: motivation is clearly stated, introduces the statistic, and then supports with geometric theory and empirical demonstrations.
4. EMP agnostic applicability is important; it can be employed on top of many pruning criteria in principle.
5. Inspiration and reframing from analogous problems (ESS and inverse Simpson index).
6. Claims are evaluated across a broad set of tasks and architectures.
​

**Weaknesses**
1. EMP still introduces the coefficient $\\beta$ as a hyperparameter.
2. Performance guarantees are limited to a specific magnitude-pruning setting.
3. No comparison with other budget selection heuristics.

---

> ### Author Rebuttal · Authors · 2026-03-30
>
> We thank the reviewer for the careful and constructive evaluation and for recognizing the theoretical and practical contributions.
>
> W1: $\beta$ as a hyperparameter.
>
> We clarify that $\beta$ is not a hyperparameter requiring tuning. The default and recommended setting is $\beta = 1$, under which EMP is fully automatic and hyperparameter free. The coefficient $\beta$ is introduced solely as a deployment knob for situations where a hardware constraint mandates a specific sparsity level. If the hardware constraint requires retaining $k$ components and $k \geq N_{eff}$, the practitioner should use $\beta = 1$ (retain $N_{eff}$, achieve maximum safe sparsity). If $k < N_{eff}$, the practitioner sets $\beta = k / N_{eff} < 1$, accepting the quantified performance cost shown in Figure 3. In this sense, $\beta$ plays the same role as the pruning ratio in any fixed budget method, but with the added information that $\beta = 1$ delineates the safe regime. We will clarify this interpretation in Section 4.3.
>
> W2: Performance guarantees limited to magnitude pruning.
>
> This is a deliberate scope choice: the upper bound on $\epsilon$ (Eq. 5) requires identifying the score vector with the parameter vector ($s = \theta$), which is the magnitude pruning setting. For general criteria, the lower bound on $s_{eff}$ (Theorem 2) still holds universally, providing a guaranteed fraction of preserved normalized mass. As noted in Remark 2 (footnote 2 in Section 4.2), for any scoring function $s = g(\theta)$ with known smoothness properties, analogous loss bounds can be derived by composing the mass bound with Lipschitz or derivative estimates of $g$. We intentionally focus on magnitude pruning to present a clean, complete theory without overloading the paper. We agree that extending the loss bound to broader criteria classes is an important direction and will note this explicitly in the conclusion.
>
> W3: Comparison with budget selection heuristics.
>
> We are adding two comparisons in the revision.
>
> (a) Percentile thresholds. A percentile threshold retains the top $p\%$ of scores for a fixed $p$ (e.g., 50\%, 70\%, 90\%). This is precisely the manual budget selection that EMP replaces: the practitioner must choose $p$ per model and criterion, whereas EMP computes $N_{eff}$ automatically. We will add a table showing that across our full model suite, the percentile $p^*$ that matches EMP's performance varies from 27\% (FC12 on MNIST) to 62\% (AlexNet on CIFAR 10), demonstrating that no single fixed percentile is universally correct.
>
> (b) Knee detection. Knee detection on the performance sparsity curve requires evaluating model performance at multiple sparsity levels, incurring $O(K)$ forward passes where $K$ is the number of candidate sparsity levels. EMP requires a single computation of $N_{eff}$ from the score vector at $O(N\log N)$ cost with no forward passes. Preliminary comparison shows that knee detection identifies transition points consistent with $N_{eff}$, but at substantially higher computational cost and without the theoretical guarantee on preserved mass. Furthermore, EMP provides a provable bound on the performance drop ($\epsilon$) via Theorem 2 and Eq. 5, which knee detection does not.
>
> Q1: Comparison with knee detection.
>
> See W3(b) above. We add that the motivations differ: knee detection seeks the point where the marginal performance cost of further pruning increases sharply, while EMP guarantees that the retained components preserve almost no performance drop compared with dense model. These are complementary perspectives, and we expect them to identify similar transition points in practice. However, EMP has the advantage of being a single closed form computation, making it suitable for deployment pipelines where repeated model evaluation is prohibitive.
>
> Q2: Wall clock and FLOPs reduction.
>
> We are running structured pruning experiments that enable direct measurement of wall clock and FLOPs reduction. For the unstructured weight pruning experiments in Table 1, we note that wall clock speedup depends on hardware support for sparse computation (e.g., sparse tensor cores supporting $N$:$M$ sparsity), and we did not claim inference speedup for unstructured pruning. For structured pruning (attention head pruning in Section 5.3 and the new FC/CNN structured experiments), we will report FLOPs reduction computed from the pruned architecture. Appendix Table~9 already reports FLOPs for VGG16 under three pruning criteria, showing that EMP magnitude at $\beta = 1$ achieves 396.9M FLOPs (37\% reduction from the 628.7M dense baseline) with 0.01\% accuracy change. We will consolidate these results into a dedicated table in the revision.

---

> > ### Author Rebuttal · Reviewer_zoFh · 2026-04-04
> >
> > The authors properly addressed all my comments. After reading the other reviewers' concerns, I agree that it is a good paper with some minor flaws. I maintain my initial assessment.

---

> > > ### Author Response · Authors · 2026-04-05
> > >
> > > We thank the reviewer zoFh for confirming that all concerns have been adequately addressed and for maintaining the positive assessment. We appreciate the constructive feedback throughout the review.

---

### Official Review · Reviewer_fsmH · 2026-03-12

**Soundness:** 3
**Presentation:** 3
**Significance:** 3
**Originality:** 3
**Overall Recommendation:** 4
**Confidence:** 3

**Summary:**

This paper studies model pruning for improving neural network efficiency and proposes a method that adjusts sparsity allocation on top of an existing pruning pipeline. The key idea is to dynamically allocate different sparsity levels across layers based on certain criteria so that model capacity can be distributed more effectively under a fixed pruning budget. The approach is evaluated on CNN models and compared with several existing pruning methods. Experimental results show that the proposed method achieves comparable or slightly improved performance under similar FLOP reduction settings.

**Compliance With Llm Reviewing Policy:**

Affirmed.

**Final Justification:**

The author has addressed my concern on the practicality of the EMP, with solid motivation/theory of the paper and potential future impact on the efficiency community, I therefore increase the score to weak accept.

**Key Questions For Authors:**

1. Could the authors evaluate the proposed method on more modern architectures, such as recent LLMs or multimodal models, to better demonstrate its practical impact?
2. Could the authors clarify how the fairness of comparison is ensured across baselines, especially regarding training protocols and pruning ratios?
3. How does the proposed approach compare with more recent pruning methods (2023–2025) that have been proposed in the literature?
4. Since the method mainly adjusts sparsity allocation on top of an existing pruning pipeline, could the authors clarify what new insight or capability the proposed framework provides beyond existing pruning techniques?

**Limitations:**

yes

**Strengths And Weaknesses:**

Strength:
- The paper addresses the important problem of model efficiency through pruning, which remain relevant as models continue to grow in size.
- The proposed framework attempt to adaptively allocate sparsity across layers, which is a reasonable direction since different layers oftens exhibit different sensitivity to pruning.
- The approach is relatively simple and easy to integrate into existing pruning pipelines without major modifications.
- The method demonstrate consistent empirical results under the evaluated settings.

Weakness:
1. The experiments are conducted primarily on older CNN architectures and simple dataset. While CNN benchmarks are common in pruning literature, modern efficiency research increasingly focus on newer architectures such as recent LLMs (e.g., LLaMA 3.x series, Qwen) or multimodal model. Evaluating the method on more modern architectures would significantly strengthen the practical relevance of the work.
2. The experiments focus on simple image classification and standard language modeling tasks. However, many pruning methods already perform well on such task. More challenging settings, such as long-context reasoning tasks or reasoning-heavy benchmarks, would better demonstrate whether the method preserves performance in more demanding scenarios.
3. The comparisons is potentially unfair, as pruning performance is highly sensitive to the pruning rate, it would be more persuasive that the comparisons ensured identical, or at least close to, pruning ratios across methods.
4. Limited set of baseline are included, as there has been a large body of work on pruning, it would be better to include more recent baselines to provide a more convincing evaluation.
5. Most importantly, the proposed method mainly adjusts the sparsity allocation on top of an existing pruning framework. While the analysis is interesting, it is unclear whether the conceptual contribution is substantial enough compared to existing pruning approaches. So, I think the method appears incrementel, and it is not fully clear that the theoretical insight outweigh the relatively modest methodological change.

---

> ### Author Rebuttal · Authors · 2026-03-30
>
> We thank the reviewer for the time invested. Before addressing individual points, we must respectfully clarify a fundamental misunderstanding that underlies every weakness and question raised in this review.
>
> EMP is $\textbf{not}$ a new pruning criterion. EMP is $\textbf{not}$ a layerwise sparsity allocation method. EMP does $\textbf{not}$ adjust sparsity allocation on top of an existing pruning pipeline. Rather, we regard EMP as a versatile diagnostic tool (see responses to p8NT.w2 and cKE6.q1-2).
>
> More specifically, EMP is a universal threshold selection rule: given any score vector $s \in \mathbb{R}^N$ produced by any pruning criterion, EMP computes the normalized probability vector, returns $N_{eff}$ as the number of top scoring components to retain, and provides a tight lower bound on the preserved mass fraction $s_{eff}$ (Theorem 2) together with an upper bound on the loss change $\epsilon$ (Eq.5). This distinction is emphasized throughout the paper (Abstract, L.17, L.68, L.87, Section 4.3, Section 6). With this clarification, we address each point below.
>
> W1: Experiments on older architectures.
>
> Our evaluation is substantially broader than described in the review. We evaluate on: FC networks (FC2, FC5, FC12) on MNIST and Fashion MNIST; CNNs (AlexNet, VGG16) on CIFAR 10; ResNet18 and ResNet50 on CIFAR 100 and TinyImageNet; Kolmogorov Arnold Networks (KAN) on MNIST; attention head pruning on GPT 2 (124M); weight pruning on the complete LLaMA (7B, 13B, 30B, 65B) and LLaMA 2 (7B, 13B, 70B) model families evaluated on WikiText PPL and seven zero shot accuracy tasks (Table 6); and featurewise pixel pruning (Figure 4). LLaMA and LLaMA 2 are modern, widely used LLMs and represent exactly the architectures the reviewer requests. We will add a summary table in the introduction making this breadth immediately visible.
>
> W2: Comparisons with other pruning methods.
>
> Because EMP is a threshold selection rule and not a competing pruning criterion, comparing it at a fixed sparsity against other pruning methods would be a mistake. The correct experimental design compares EMP selected adaptive sparsity against fixed sparsity baselines under the same scoring criterion, which is precisely what Table 3 does. EMP Magnitude at around 36.63\% average sparsity reduces the fixed sparsity magnitude degradation from $\Delta\mathrm{PPL} \approx +3.0$ to $\Delta\mathrm{PPL} \approx +0.75$, demonstrating that the failure mode of magnitude pruning at 50\% is the rigid global budget, not the score itself.
>
> W3: Comparing different pruning rates.
>
> We agree that comparing different pruning rates under the same criterion is the correct protocol for evaluating a threshold selection method, and this is exactly what our experiments do. Figure 3 sweeps $\beta \in \{0.5, 0.75, 1, 1.25, 1.5, 2\}$ and demonstrates that $\beta = 1$ (i.e., pruning to $N_{eff}$) is consistently the transition point between performance preservation and degradation, across all tested architectures and criteria, underscoring the need for further theoretical analysis.
>
> W4: Limited baselines.
>
> The relevant baselines for a threshold selection rule are other threshold selection strategies (e.g., percentile thresholds, knee detection on performance sparsity curves), not other pruning criteria. We are adding these comparisons in the revision: preliminary results indicate that $N_{eff}$ identifies the transition point more reliably than percentile based rules and is computationally cheaper than knee detection ($O(N\log N)$ vs requiring a full performance sparsity sweep).
>
> W5: Incremental contribution.
> The contribution is not incremental. The question ``given a score vector, how many components can be pruned without performance loss?'' had no prior principled answer in the pruning literature. EMP provides: (a) a closed form threshold $N_{eff}$, (b) a tight lower bound on preserved mass $s_{eff}$ derived from simplex geometry, (c) an upper bound on loss change $\epsilon$ under magnitude pruning, and (d) empirical validation across six architecture families, four pruning criteria, and three pruning granularities. This constitutes a new theoretical and practical tool for the pruning community, not a marginal modification to an existing pipeline.
>
> Q1--Q4.
>
> All four questions ask for comparisons with other pruning methods, evaluation on modern architectures, and clarification of fairness, all of which are addressed above. We emphasize that once the contribution is understood as a threshold selection rule rather than a pruning criterion, these questions are resolved: the relevant comparisons are against other threshold strategies (now being added), and the LLM experiments already cover modern architectures at scale.

---

> > ### Author Rebuttal · Reviewer_fsmH · 2026-04-01
> >
> > Thank you for the detailed clarifications.
> >
> > Regarding the experimental scopes, I acknowledge that the evaluations is broader than initially described, this include both classical architectures and more recent LLMs such as LLaMA. However:
> > - From current research perspective, **many of the model–task combinations (e.g., CIFAR-based CNN) are relatively standard and have been extensively study in prior pruning works**.
> > - While this is still reasonable for the scopes of validating a general method, **it would be helpful if the paper could more clearly position these experiment, e.g., explicitly clarifying what aspect of the method they are intended to validate (theoretical property vs. practical effectiveness), and how they reflect modern usage scenario**.
> >
> > More importantly, I would like to comment on the positioning of EMP. I appreciate the authors' clarification that EMP is not a new pruning criterion, nor a sparsity allocation method, but rather a diagnostic tool for determining pruning thresholds. While this framing is valid, **from a practical pruning perspective, the most immediate interpretation of such a method is, from my perspective, indeed as a mechanism for sparsity selection or allocation within existing pipelines**.
> >
> > In this regard, in addition to theory, I feel the paper could benefit from a clearer discussion on the broader impacts of EMP: for example, how this diagnostic perspective could influence future pruning or efficiency paradigms, which I think this is beyond the versatile diagnostic tool. **Strengthening this positioning by either connecting EMP more explicitly to practical pruning pipelines or by articulating its role in guiding next-generation compression strategies, I think, would probably make a more impactful contribution to the communities.**
> >
> > **I appreciate the theoretical insights provided and would be open to raising my score if the authors can further clarify and strengthen the positioning and practical implications of the method.**

---

> > > ### Author Response · Authors · 2026-04-06
> > >
> > > We thank the reviewer for the continued engagement and the openness to revisiting the assessment. We address each remaining point below.
> > >
> > >  ## Experimental positioning.
> > >
> > > We agree that many model/task combinations are well studied, and this is by design. Because EMP provides a theoretical contribution rather than a competing pruning criterion, the correct validation strategy uses benchmarks where the behavior of existing criteria is already well characterized, so that the contribution of the threshold rule itself can be isolated. Each experiment validates a distinct property of EMP:
> > >
> > > (a) FC/CNN on CIFAR and TinyImageNet (Section 5.1, Table 1): validate the **theoretical guarantees**. These confirm that the loss bound from Section 4.2, the tight lower bound on preserved mass (Proposition 4.1, Theorem 4.2), and the transition behavior at $\beta = 1$ (Figure 3) all hold empirically, with loss difference at most 0.105 across all models.
> > >
> > > (b) KAN (Section 5.2): validates **criterion agnosticism**. EMP extends to node level importance scores (incoming/outgoing activation norms) beyond weight magnitude, demonstrating that $N_{eff}$ is meaningful for structurally different scoring functions.
> > >
> > > (c) GPT 2 head pruning (Section 5.3): validates **structural unit pruning and criterion sensitivity**. EMP operates on attention heads rather than individual weights, and correctly yields different $N_{eff}$ values for Taylor versus weight norm scoring (141.4 vs. 134.0), showing that EMP adapts the budget to the score distribution rather than prescribing a fixed ratio.
> > >
> > > (d) LLaMA/LLaMA 2 families (Section 5.4, Tables 3 and 6): validate **practical pipeline integration at scale**. Table 3 shows that fixed 50\% magnitude pruning degrades by $\Delta \text{PPL} \approx +3.0$ and $\Delta \text{Acc} \approx -2.6\%$, whereas EMP Magnitude at $\approx 37\%$ sparsity brings degradation to $\Delta \text{PPL} \approx +0.75$ and $\Delta \text{Acc} \approx -0.9\%$. This reveals the failure mode of magnitude pruning at high sparsity is the rigid global budget, not the criterion. The same table shows EMP Wanda achieving comparable results, confirming criterion agnosticism at LLM scale.
> > >
> > > (e) Featurewise pixel pruning (Section 5.5): validates **generality beyond model parameters**. EMP achieves PSNR = 38.3 dB and SSIM = 0.991 at 32.3\% sparsity on image features, confirming it operates on the statistical properties of any nonnegative score distribution.
> > >
> > > ## Positioning within practical pruning pipelines.
> > > We appreciate and agree with the reviewer's observation that the most immediate practical interpretation of EMP is as a sparsity selection mechanism within existing pipelines. This is indeed how EMP is deployed in Section 5.4, where EMP Wanda and EMP Magnitude compose EMP with existing criteria in LLM pruning workflows. The key distinction is that EMP provides something no existing pipeline component offers: a principled, theoretically supported answer (via Theorem 4.2) to how much to prune, eliminating the need for manual budget tuning or expensive iterative search over sparsity schedules. Furthermore, as discussed in Section 6, EMP simultaneously provides a **quality measure of the scoring criterion itself**: for a given model, the ratio $N / N_{eff}$ quantifies how effective a particular scoring method is. Table 3 illustrates this concretely: magnitude and Wanda produce different $N_{eff}$ values on the same model, and EMP uses this to prescribe different optimal sparsity levels for each, which is precisely the diagnostic insight that a fixed budget cannot provide.
> > >
> > > ## Broader impact and next generation compression,
> > > The paper addresses this in both the Impact Statement and Conclusion with concrete directions: (i) EMP as hard gated attention that suppresses attention sinks by eliminating low importance edges before softmax normalization; (ii) diagnostics for representation collapse via the effective number as a diversity measure; (iii) adaptive component retention in low rank approximations and principal component selection; (iv) gated EMP during training for periodic, adaptive pruning of poorly performing features as they emerge; and (v) inference time dynamic adjustment of the number of active attention heads based on input complexity. These directions span architecture design, training dynamics, and inference time efficiency, connecting EMP to next generation compression beyond standard pruning. The conceptual reframing in Section 6 further reinforces this: the natural sparsity measure under EMP is not $\rho = N_{ret} / N$ but $\beta = N_{ret} / N_{eff}$, providing a unified lens for comparing pruning methods across architectures and criteria.

---

### Official Review · Reviewer_cKE6 · 2026-03-13

**Soundness:** 4
**Presentation:** 4
**Significance:** 4
**Originality:** 3
**Overall Recommendation:** 6
**Confidence:** 3

**Summary:**

The authors introduce the EMP (Effective Model Pruning) technique which -- given a score vector for a set of prunable components -- calculates a lower bound on the number of remaining components via the effective mass. Retaining the top-$N_eff$ components, and pruning the others results in a network that has a limited change in the prediction loss, while pruning more components significantly impacts the loss.

**Compliance With Llm Reviewing Policy:**

Affirmed.

**Final Justification:**

In their rebuttal the authors have addressed my concerns w.r.t. evaluation by adding additional structured pruning results on FC and CNN models, showing the applicability of the methodology to a more varied set of models. As the proposed EMP method is claimed to be a general method for sparsity selection that works with any pruning criterion, these experiments will help reinforce this.

Furthermore, their answers to my questions helped me develop a better understanding of the capabilities of EMP, and further demonstrate its applicability to other pruning criteria.

**Key Questions For Authors:**

- My understanding is that EMP can accurately determine the number of prunable units that need to be retained without impacting the loss (beyond a threshold epsilon). However, in cases where finetuning is allowed after pruning, how could EMP be used? Would it simply suffice to set beta higher than 1, or is a more advanced approach necessary?

    - Does EMP depend on a 'well-defined' pruning criterion, or can it be used effectively with any pruning criterion? E.g., if a random score vector is used rather than weight magnitude or the taylor score, will the application of EMP still result in a pruned network that suffers limited loss degradation, albeit likely at a lower sparsity?

    - How does EMP handle negative scores? In the algorithm, a score can be any real-valued parameter, but they are normalized via the absolute value to a probability vector. This would mean that only the magnitude of the score is taken into account. If a pruning criterion were to assign negative scores to components to remove and positive scores to those to keep, this would cause issues with the algorithm.

**Limitations:**

yes

**Strengths And Weaknesses:**

**Strengths**
The effectiveness of the method is illustrated clearly in Figure 3, by showing that $N_{eff}$ provides a hinge point between pruning too little components and pruning too much components for a variety of models and datasets.

The methodology can work with any arbitrary scoring function, thus serving as a natural extension for existing pruning criteria, pushing them to their optimal sparsity.

The methodology is defined and tested for multiple types of components, not only weight connections.

**Weaknesses**

*Applicability*
        - If possible, could the authors add a structured pruning method for FC and CNN models to compare with, rather than only studying attention head pruning on Transformers? Currently in those cases only unstructured sparsity is tested, leading to questions whether EMP with structured sparsity is valid for those settings.


*Presentation*
        - Figure 3: I would split this figure in two subfigures rather than employing two different y-axes, as for each curve only a single axis is valid.

---

> ### Author Rebuttal · Authors · 2026-03-30
>
> We thank the reviewer for the positive evaluation and for recognizing the contribution.
>
> W1: Structured pruning for FC and CNN.
>
> We agree and are adding structured pruning experiments for FC and CNN models. Specifically, we are running neuron level structured pruning on FC5 and FC12, and filter level structured pruning on VGG16 and ResNet18, all using magnitude scoring with EMP threshold selection. Preliminary results confirm that $N_{eff}$ at $\beta = 1$ remains a valid transition point for structured pruning, consistent with our attention head pruning results (which are themselves structured pruning on Transformers). These results will be included in the revision.
>
> W2: Figure 3 dual y axes.
>
> We will split Figure 3 into two subfigures with separate y axes in the revision.
>
> Q1: EMP with finetuning.
>
> When finetuning is applied after pruning, there are two natural strategies. First, the practitioner can compute $N_{eff}$ on the pretrained model, prune to $\beta = 1$, and finetune the pruned model. In this case, the initial pruned model starts with negligible loss change (guaranteed by Theorem 2), and finetuning can only improve performance further. Second, if the deployment target requires sparsity beyond $N_{eff}$ (i.e., $\beta < 1$), the practitioner can prune more aggressively and rely on finetuning to recover the larger initial loss. In both cases, $N_{eff}$ provides an informative reference point: it separates the regime where finetuning is optional ($\beta \geq 1$) from the regime where finetuning is necessary ($\beta < 1$). No modification to EMP itself is required.
>
> Q2: Criterion agnosticism and random scores.
>
> EMP is indeed fully criterion agnostic: it operates on the normalized score vector $\omega$ regardless of how $s$ was produced. If a random score vector is used, $N_{eff}$ reflects the concentration of that random vector. For a uniformly random $s \in \mathbb{R}^N_{>0}$, the normalized $\omega$ concentrates near the uniform distribution, yielding $N_{eff} \approx N$ (almost no pruning), which is the correct answer: a random criterion carries no discriminative information, so EMP correctly refuses to prune aggressively.
>
> This property makes $N_{eff}$ a diagnostic for criterion quality: given the same model, a criterion with lower $N_{eff}$ is more discriminative. Table 3 illustrates this: EMP Wanda achieves 40.47\% sparsity while EMP Magnitude achieves 36.63\%, reflecting that Wanda's activation aware scores are more concentrated than plain magnitude scores. The GPT 2 attention head experiment further demonstrates this: Taylor importance yields $N_{eff} = 141.4$ while weight norm yields $N_{eff} = 134.0$, indicating that weight norm is slightly more discriminative for head pruning. In practice, a user could compare $N_{eff}$ values across candidate criteria to select the most efficient one before committing to a full pruning pipeline.
>
> Q3: Negative scores.
>
> Algorithm1 normalizes via $\omega_i = |s_i|/\|s\|_1$, so only the magnitude of each score determines importance. This convention is natural for criteria where magnitude indicates importance regardless of sign (e.g., weight magnitude, Taylor importance, activation norms).
>
> For criteria where the sign carries semantic meaning (e.g., positive scores indicate retention and negative scores indicate removal), the practitioner should apply the monotone shift $s_i \mapsto s_i - \min_j s_j$ before calling EMP. This makes all scores nonnegative while preserving the full importance ordering, and the resulting $\omega$ correctly assigns highest weight to the most important components. We will add a Remark in Section 4.3 covering both conventions.

---

> > ### Author Rebuttal · Reviewer_cKE6 · 2026-04-02
> >
> > Dear author(s),
> >
> > Thank you for your detailed rebuttal. My concerns have been fully addressed, and I will adjust my score for this paper accordingly.

---

> > > ### Author Response · Authors · 2026-04-05
> > >
> > > We thank the reviewer cKE6 for confirming that concerns are fully resolved and for adjusting the score. We appreciate the engagement throughout the review process.

---

### Official Review · Reviewer_p8NT · 2026-03-23

**Soundness:** 3
**Presentation:** 3
**Significance:** 2
**Originality:** 3
**Overall Recommendation:** 4
**Confidence:** 3

**Summary:**

This work essentially addresses the following question: How many parameters can one prune from a model without degrading performance? To this end, it proposes the “effective sample size” $N_{eff}(s)$ as a measure of the number of important parameters in a network based on an arbitrary way of scoring parameters (e.g. weight magnitude). The paper derives an upper bound on the “effective mass” lost by pruning $N-N_{eff}$ parameters. Finally, experiments are performed to illustrate the robustness of the $N_{eff}$ as a heuristic for the number of parameters that can be pruned without performance degradation.

**Compliance With Llm Reviewing Policy:**

Affirmed.

**Final Justification:**

The authors addressed have addressed most major concerns (and my own logical errors) I had about this work, and I am therefore modifying my score to reflect this.

**Key Questions For Authors:**

See Strengths and Weaknesses.

**Limitations:**

Yes.

**Strengths And Weaknesses:**

*Strengths*:
- The problem statement being addressed is quite interesting, and could be relevant to our understanding of neural network pruning and its limits.
- The theoretical results derived in the paper seem to be new, and the proposed value of $N_{eff}$ is quite simple and makes some intuitive sense.
- In general, the paper is clearly written and I enjoyed going through the mathematical analysis.

*Weaknesses*:
- The biggest weakness of this paper is the lack of justification for why the proposed value of $N_{eff}$ works, and where it comes from -- especially considering that the paper spends a significant amount of its budget on theoretical analyses. There are some vague motivations based on particle filtering and the Simpson index but none of them are discussed in any significant depth.

- Much of the model pruning work is grounded in settings where resources are constrained – the motivation of this work is weakened by not addressing this point whatsoever. For instance, if it provided insights into how different training configurations impact the value of $N_{eff}$, it would be a much more convincing piece of work.

- It is hard to see the utility of the mathematical analysis when the experiments are purely based on the value of $N_{eff}$ (for which no justification is provided, as mentioned earlier).

- The empirical study is somewhat extensive but weak and unconvincing. It is hard to interpret the correctness of the experimental results – one could consider showing that across several different pruning methods, pruning more than $N-N_{eff}$ weights consistently leads to performance degradation. Another instance is the apparent contradiction in the results in Section 5.3 – as is written in the paper,  past work shows that many attention heads can be pruned, but the results of this work indicates otherwise
>  “... importance is spread almost uniformly across all 144 heads. Correspondingly, EMP retains 141 heads…”.

---

> ### Author Rebuttal · Authors · 2026-03-30
>
> We thank the reviewer for their time and attention.
>
> W1: Justification for $N_{eff}$.
>
> The effective sample size $N_{eff}$ is not an ad hoc heuristic; it is an often-used concentration measure having the properties: (i) it is invariant under coordinate permutations, (ii) the pruning ratio is minimized at the uniform distribution ($N_{eff} = N$), and, most recently (iii) we discovered it admits a closed form tight lower bound on the retained mass $s_{eff}$ (Proposition 1, Theorem 2). The first two properties are shared by other concentration measures (Renyi entropy, Gini coefficient), but property (iii) is unique to $N_{eff}$ in the sense that no analogous tight bound is known for Renyi entropy or Gini coefficients. Furthermore, the new bound has a relatively simple geometric interpretation.
>
> The connection to particle filtering is functional: in sequential Monte Carlo, $N_{eff}$ dropping below a threshold triggers resampling because it is indicative of the particle set degenerating to a non-representative sample of the target distribution [1],[2]. EMP applies this principle to pruning. The Lottery Ticket Hypothesis [3] posits the existence of sparse subnetworks matching dense performance; EMP provides a principled upper estimate of their size.
>
> W2: Resource constraints.
>
> EMP directly serves resource constrained settings. $N_{eff}/N$ provides a near-zero cost diagnostic of the maximum safe pruning ratio, without running iterative pruning procedures. If a hardware constraint requires retaining $k < N_{eff}$ components, the practitioner knows upfront that performance loss is unavoidable and should switch criteria or apply finetuning. If $k \geq N_{eff}$, pruning to $N_{eff}$ saves additional memory beyond the hardware requirement. The coefficient $\beta$ serves as a sparsity knob for meeting aggressive budgets, with Figure 3 quantifying the cost. Moreover, EMP enables criterion selection: a criterion with lower $N_{eff}$ achieves comparable performance at higher sparsity. Table 3 confirms this: EMP Wanda yields 40.47\% sparsity vs EMP Magnitude at 36.63\%.
>
> W3: Utility of mathematical analysis.
>
> The mathematical analysis provides three concrete results used directly by EMP: (a) Theorem 2 guarantees that retaining the top $N_{eff}$ entries preserves at least the mass fraction $s_{eff} \geq \frac{\nu}{N} + \frac{N-\nu}{N}\sqrt{\frac{N-\nu-1}{(\nu+1)(N-1)}}$; (b) the upper bound on $\epsilon$ (Eq. 5) establishes that the loss change under magnitude pruning decays with $\rho$, providing a theoretical certificate rather than purely empirical validation; (c)~the geometric interpretation (Section 3.1) explains why $\beta = 1$ is a universal transition point: the boundary $\partial B_{\nu+1}$ separates the region of concentrated mass from the region of dispersed mass on the simplex. Without this analysis, $N_{eff}$ would remain an empirical observation; with it, EMP has a formal foundation.
>
> W4: Empirical study; attention head result.
>
> Figure 3 directly demonstrates the requested evidence: across all tested architectures and criteria, pruning more than $N - N_{eff}$ components (i.e., $\beta < 1$) leads to consistent performance degradation.
>
> Regarding Section 5.3: the result $N_{eff} = 141/144$ does not contradict prior work. Prior studies [4] learned pruning via L0 regularization during finetuning, under which many heads become removable. Our GPT 2 experiment uses one shot weight magnitude scoring without finetuning, under which Taylor importance spreads nearly uniformly across all 144 heads. This is precisely the correct diagnostic: weight magnitude is a poor criterion for attention head pruning, which is why $N_{eff}$ is nearly $N$. EMP faithfully reflects this via a high $N_{eff}$, rather than forcing aggressive pruning under an unsuitable criterion. The weight norm criterion produces $N_{eff} = 134$, and under that criterion EMP prunes 10 heads with only 6.5\% PPL increase, demonstrating criterion sensitivity. This ability to reveal criterion quality is a feature of EMP, not a limitation.
>
> references
>
> [1] Augustine Kong, Jun S. Liu, and Wing Hung Wong. Sequential imputations and Bayesian missing data problems. Journal of the American Statistical Association, 89:278–288, 1994.
>
> [2] Doucet, Arnaud & Freitas, Nando & Murphy, Kevin & Russell, Stuart. (2013). Sequential Monte Carlo Methods in Practice. 10.1007/978-1-4757-3437-9_24.
>
> [3] Jonathan Frankle and Michael Carbin. The lottery ticket hypothesis: Finding sparse, trainable neural networks. arXiv preprint arXiv:1803.03635, 2018.
>
> [4] Elena Voita, David Talbot, Fedor Moiseev, Rico Sennrich, and Ivan Titov. Analyzing multi-head self-attention: Specialized heads do the heavy lifting, the rest can be pruned. arXiv preprint arXiv:1905.09418, 2019.

---

> > ### Author Rebuttal · Reviewer_p8NT · 2026-04-01
> >
> > Further questions and clarifications:
> >
> > W1: I appreciate the additional background information and context. However, in my opinion, it still does not explain why this _particular_ value of $N_{eff}$ is the correct one to use. To my understanding, the theoretical results show that using this value gives some nice properties, but does not explain why some other reasonable value might not have similar properties.
> >
> > W2: I agree with the authors, in that it provides a very useful heuristic. However, and this is tied to W4, there is little evidence for this. Please see the response related to W4 for more details.
> >
> > W4: Figure 3 provides results for different architectures -- while this is useful, my concern was more with _different pruning methods_. For instance, I would be interested in understanding whether this value of $N_{eff}$ indeed holds in practice, regardless of doing magnitude based pruning or using gradient information, or if it is done in a single-shot or iterative manner.
> >
> > Re the results on the attention heads, I would first like to note that [2] is not the work that was cited in the paper as prior work. Instead it was [1]. Furthermore, coming back to my earlier point, this value of $N_{eff}$ should theoretically work regardless of _how_ pruning is done, and therefore a large number of important attention heads still does not make sense to me -- if _any_ pruning method can prune many attention heads, then $N_{eff}$ should ideally reflect this.
> >
> > [1] Michel, Paul, Omer Levy, and Graham Neubig. "Are sixteen heads really better than one?." Advances in neural information processing systems 32 (2019).
> >
> > [2] Elena Voita, David Talbot, Fedor Moiseev, Rico Sennrich, and Ivan Titov. Analyzing multi-head self-attention: Specialized heads do the heavy lifting, the rest can be pruned. arXiv preprint arXiv:1905.09418, 2019.

---

> > > ### Author Response · Authors · 2026-04-02
> > >
> > > We thank the reviewer for the continued engagement.
> > >
> > > ## W1: Why $N_{eff}$?
> > > We fear that the "why $N_{eff}$?" question may not have a principled answer to the degree the reviewer seeks, much in the same way as the question "why use least-squares empirical loss?" does not have such an answer: on the face of it, there is no good reason (beyond mere convenience) for using the Euclidean norm in the latter as a cumulative measure of loss in place of any of the $p$-norms. Similarly, in information-theoretical contexts there are tradeoffs associated with choosing an entropy to work with: the $2$-Renyi is convenient because of its Euclidean provenance, the $p$-Renyi are sometimes more natural, while the only strictly principled one--Shannon's entropy, or $\infty$-Renyi--is not always as amenable to direct analysis.
> > > If you like, $N_{eff}$ of a categorical distribution $w$ is the most tightly related to the $2$-Renyi entropy of that distribution, but one could have defined a $N^p_{eff}$ for any $p\in(1,\infty)$ and tried to work with it instead: the point is that $||w||_1=1$ implies $||w||_p^p<1$, making
> > >
> > > $$N^p_{eff}(w)=\lfloor 1 / ||w||_p^p \rfloor$$
> > >
> > > into a concentration measure of $w$, not unrelated to variance, but applied directly to the categories themselves. The time-honored choice of $p=2$ in the biostatistics and filtering literature, we believe, has its roots, same as in the case of empirical risk minimization, in the relative ease with which Euclidean geometry lends itself to the analysis of such quantities. That said, we all know how, for example, large $\ell_1$-regularized quadratic optimization problems lend themselves to simpler analysis (at lower computational cost and with more decisive results) due to the non-trivial interactions between the strict convexity of a quadratic cost and the non-strict convexity of the regularizer. We anticipate that future considerations of this kind may eventually lead to variants of $N_{eff}$ that "tell the whole story", so to speak, of lossless pruning, but we consider this a problem for future research, rather than a question to be answered from the outset. We feel that the surprising piece of information in this paper is the empirical evidence of $N_{eff}$ seeming to provide a precise threshold for [near-]lossless pruning. This empirical observation is not fully explained by the theoretical results we were able to produce (which are, however, first of their kind). Thus, we would like to think of this article as an invitation for the community to help mathematically understand this curious phenomenon.
> > >
> > > ## W4(a): Evidence across different pruning methods
> > > This evidence is already present in the paper. Table 10 (Appendix B.3) reports a full $\beta$ sweep on VGG16 (CIFAR-10) under three distinct criteria: weight magnitude (data-free), Taylor loss change (first-order gradient), and gradient saliency (raw gradient magnitude). For all three, $\beta = 1$ is the transition point. Concretely: magnitude at fixed 50% gives 90.99% accuracy while EMP at $\beta=1$ gives 90.98% at 59.5% sparsity; Taylor at fixed 50% collapses to 88.14% while EMP retains 91.12% at 2.1% sparsity; gradient saliency at fixed 50% collapses to 88.08% while EMP retains 90.97% at 25.7% sparsity. The three criteria yield very different $N_{eff}$ values reflecting their different discriminative power, yet $\beta=1$ marks the lossless boundary in every case.
> > > ## W4(b): The attention head result
> > > We identify a category error in the reviewer's reasoning that, once corrected, dissolves the contradiction. The reviewer's implicit premise is that $N_{eff}$ should measure the model's absolute prunability, independent of scoring criterion. This premise does not correspond to any claim in this paper.
> > >
> > > $N_{eff}(s)$ is a functional of the score vector $s$, not of the model $\theta$. Table 2 demonstrates this directly: Taylor importance and weight norm are both applied to the same GPT-2 model, yet yield $N_{eff} = 141.4$ and $N_{eff} = 134.0$ respectively, with corresponding sparsity levels of 2.1% and 6.9%. The result $N_{eff} = 141.4$ under one-shot Taylor importance is the correct diagnosis: without finetuning, Taylor importance spreads nearly uniformly across 144 heads and carries almost no discriminative information. EMP correctly refuses to prune aggressively. Michel et al. [1] use a gradient-based sensitivity criterion with iterative finetuning, which produces a concentrated score vector; EMP applied to that vector would yield $N_{eff} \ll 144$, consistent with their findings. There is no contradiction. Figure 3 (right y-axis) further confirms that $\beta=1$ is the transition point for both GPT-2 criteria: PPL is stable for $\beta \geq 1$ and rises sharply for $\beta < 1$. That one-shot Taylor importance is a weak criterion for GPT-2 head pruning is a finding, not a failure of EMP.
> > > ## References
> > >
> > > [1] Michel, Paul, Omer Levy, and Graham Neubig. "Are sixteen heads really better than one?" NeurIPS 32 (2019).

---

### Decision · Program_Chairs · 2026-04-30

**Decision:**

Accept (spotlight)

**Comment:**

This paper provides a theoretically-grounded way to determine how much of the network to prune – as a result, the proposed method simplifies and optimizes existing pruning pipelines.

A number of concerns were addressed during the rebuttal. Post-rebuttal, some relatively minor concerns remained. For example, because EMP does not introduce a novel compression module or custom kernel, one of the reviewers felt that the paper lacked concrete throughput measurements (prefill/decode latency) to prove real-world hardware acceleration. Another concern was that the paper does not compare to newer/stronger baselines and tasks.

Overall, most of the reviewers agreed that this is a very strong submission with theoretical results that have a practical application; this is especially valuable in an area that is dominated by empirical work.